



# A new approach to simulate peat accumulation, degradation and stability in a global land surface scheme (JULES vn5.8_accumulate_soil)

Sarah E. Chadburn[1], Eleanor J. Burke[2], Angela V. Gallego-Sala[3], Noah D. Smith[1], M. Syndonia Bret-Harte[4], Dan J. Charman[3], Julia Drewer[5], Colin W. Edgar[4], Eugenie S. Euskirchen[4], Krzysztof Fortuniak[6], Yao Gao[7], Mahdi Nakhavali[3], Włodzimierz Pawlak[6], Edward A.G. Schuur[8], and Sebastian Westermann[9]

[1]Department of Mathematics, University of Exeter, Exeter, UK
[2]Met Office Hadley Centre, Exeter, UK
[3]Geography Department, University of Exeter, Exeter, UK
[4]Institute of Arctic Biology, University of Alaska Fairbanks, USA
[5]UK Centre for Ecology & Hydrology, Bush Estate, Penicuik, Scotland, UK
[6]Department of Meteorology and Climatology, University of Lodz, Poland
[7]Finnish Meteorological Institute, Helsinki, Finland
[8]Center for Ecosystem Science and Society, Northern Arizona University, Flagstaff, USA
[9]Department of Geosciences, University of Oslo, Oslo, Norway

**Correspondence:** Sarah Chadburn (s.e.chadburn@exeter.ac.uk)

**Abstract.**

Peatlands have often been neglected in Earth System Models (ESMs). Where they are included, they are usually represented via a separate, prescribed grid cell fraction that is given the physical characteristics of a peat (highly organic) soil. However, in reality soils vary on a spectrum between purely mineral soil (no organic material), and purely organic soil, typically with an

organic layer of variable thickness overlying mineral soil below. They are also dynamic, with organic layer thickness and its properties changing over time. Neither the spectrum of soil types nor their dynamic nature can be captured by current ESMs.

Here we present a new version of an ESM land surface scheme (Joint UK Land Environment Simulator, JULES) where soil organic matter accumulation – and thus peatland formation, degradation and stability – is integrated in the vertically-resolved soil carbon scheme. We also introduce the capacity to track soil carbon age as a function of depth in JULES, and compare this

to measured peat age-depth profiles.

This scheme simulates dynamic feedbacks between the soil organic material and its thermal and hydraulic characteristics. We show that draining the peatlands can lead to significant carbon loss along with soil compaction and changes in peat properties. However, negative feedbacks can lead to the potential for peatlands to rewet themselves following drainage. These ecohydrological feedbacks can also lead to peatlands maintaining themselves in climates where peat formation would not otherwise

initiate in the model, i.e. displaying some degree of resilience.

The new model produces similar results to the original model for mineral soils, and realistic profiles of soil organic carbon for peatlands. In particular the best performing configurations had root mean squared error (RMSE) in carbon density for peat





sites of 7.7–16.7 kgC m$^{-3}$ depending on climate zone, when compared against typical peat profiles based on 216 sites from a global dataset of peat cores. This error is considerably smaller than the soil carbon itself (around 30–60 kgC m$^{-3}$) and reduced

by 35–80% compared with standard JULES. The RMSE at mineral soil sites is also smaller in JULES-Peat than JULES itself (reduced by ∼30–50%). Thus JULES-Peat can be used as a complete scheme that simulates both organic and mineral soils. It does not require any additional input data and introduces minimal additional variables to the model. This provides a new approach for improving the simulation of organic and peatland soils, and associated carbon-cycle feedbacks in ESMs, which other land surface models could follow.

## 1 Introduction

Peatlands are extremely carbon-dense ecosystems, occupying only around 3% of the land surface but storing up to 30% of the vast soil carbon stock (Frolking et al., 2011). High latitude peatlands alone store more than 400 GtC (Hugelius et al., 2020) and tropical peatland carbon is thought to be more than 100 GtC (Dargie et al., 2017). This carbon stock has accumulated over millennia - approximately 10,000 years since the last glacial maximum - but can be released very quickly if the peatland

becomes dry or otherwise loses its function (Maljanen et al., 2010; Tiemeyer et al., 2016). This has been taking place across the world's peatlands over the last ∼ 170 years due to land use conversion for agriculture, leading to additional greenhouse gas emissions (Leifeld and Menichetti, 2018). Climate change may also lead to drying or shifts in vegetation that drive carbon loss in currently functional peatlands (Swindles et al., 2019; Dieleman et al., 2015). In addition, peat fires are increasing in severity under climate change (e.g. Scholten et al. (2021)). Thus, this carbon stock is both large and vulnerable.

It is therefore vital that we include peatlands in Earth System Models (ESMs) that are used to make projections of future climate change, including feedbacks within the global carbon cycle (Loisel et al., 2021). However, none of the models in the recent 6th Coupled Model Intercomparison Project included a representation of peatlands (Arora et al., 2020).

Peatlands can display both vulnerability and resilience via a suite of autogenic feedbacks (Waddington et al., 2015), with self-restoring properties that allow them to persist in conditions where they would not form today, but with the potential for

rapid carbon losses if they are pushed beyond their resilience threshold. In particular, the physical characteristics of peat can change over time - often in response to changes in water table or permafrost thaw (see Frolking et al. (2011)) - and, in turn, this influences the hydrological dynamics. Up to a certain point, peat that is more decomposed holds water better. Thus, if a peatland water table drops and peat starts to decompose, the peat that is more decomposed leads to increased water-holding capacity and can bring the water table back up again, leading to resilience. On the other hand, if the peat drainage or decomposition

is more severe, it can cross a threshold where it loses the ability to maintain its water table, leading to rapid carbon loss and further degradation of the soil structure. This threshold is shown for example in Wang et al. (2021) Figure 7, where the soil characteristics change dramatically above a threshold bulk density of 200 kgm$^{-3}$ (0.2 g cm$^{-3}$).

The global land surface schemes that do simulate peatland carbon stocks (Qiu et al., 2018; Bechtold et al., 2019; Müller and Joos, 2020) do not simulate the interplay of processes that leads to the self-sustaining and threshold-type behaviours. Thus, the

vulnerability of the carbon stocks in such models cannot be properly simulated. In particular, while modellers have prescribed





thermal and hydraulic properties for organic soils (Beringer et al., 2001; Lawrence and Slater, 2008; Chadburn et al., 2015b; Guimberteau et al., 2018), they do not let these parameters vary dynamically as the carbon in the soil changes - for example, an organic layer might decompose substantially during the course of a simulation, and therefore its thermal and hydraulic properties should also change. Occasionally models have simulated such a coupling with a limited set of parameters (Koven

et al., 2009), but none have produced a fully coupled version.

Since these dynamics are driven by changes in the vertical structure of the soil organic matter, it is important to resolve the vertical profile of soil carbon (as opposed to a scheme where the soil carbon is treated as a single 'box', e.g. JULES-CN in Wiltshire et al. (2021)). Previous studies have shown that the standard vertically-resolved soil carbon scheme in ESMs fails to recreate soil carbon profiles at sites with peat or a thick organic layer (Chadburn et al., 2017). Essentially, the models are

not able to accumulate peat on top of the soil column, since the soil layers are not allowed to grow or shrink, so carbon is continually added to the top soil layer which contains an unrealistically high carbon content, and the high carbon concentration does not extend far enough into deeper soil layers.

Specialised peat models such as DigiBog and the Holocene Peat Model (HPM) (Baird et al., 2012; Frolking et al., 2010) vertically resolve peatland structure by tracking the carbon that is added each year by treating it as a separate layer added on

top of the soil column. This results in a very large number of layers which would be computationally unwieldy for global modelling. It is also only applicable to peatlands and doesn't provide the functionality to model the continuous transitions between mineral and organic soils (both in time and space).

In this paper we present a new scheme that resolves these issues, allowing vertical accumulation of peat and dynamic coupling between thermal and hydraulic soil properties. This scheme is implemented and demonstrated in the JULES land

surface model. However, the new methods and relationships we use in this model can be used to improve other land surface schemes.

## 2   Model description

### 2.1   Overview of standard JULES

JULES is the land surface model used in the UK Earth System Model (UKESM) (Sellar et al., 2019). It is a community

model that represents the surface energy balance, heat and water fluxes, snowpack dynamics, vegetation dynamics, soil bio-geochemistry, and carbon and nitrogen fluxes (Best et al., 2011; Burke et al., 2017a; Clark et al., 2011; Harper et al., 2016; Wiltshire et al., 2021). As well as being used in UKESM, JULES takes part in multimodel analyses such as the Inter-Sectoral Model Intercomparison Project (Rosenzweig et al., 2017) and the Global Carbon Project (Friedlingstein et al., 2019; Saunois et al., 2020) and has been used to make global projections, for example, of future hydrology, permafrost thaw, and carbon and

methane emissions/climate feedbacks (Burke et al., 2017b; Chadburn et al., 2015b; Comyn-Platt et al., 2018; Gedney et al., 2019)

JULES includes a vertically-resolved soil carbon scheme (Burke et al., 2017a), although this hasn't yet been used in the Earth System Model configuration. The scheme is based on Roth-C (Jenkinson, 1990; Jenkinson and Coleman, 1999) with the





carbon pools of the Roth-C model simulated separately for each soil layer. Some vertical processes have been added such as
a diffusive mixing, which represents bioturbation and/or cryoturbation (see Burke et al. (2017a) for details). This soil carbon
scheme has more recently been coupled to a vertically-resolved nitrogen model described in Wiltshire et al. (2021). In this
paper we build on this vertically-resolved soil carbon-nitrogen scheme in JULES.

## 2.2 Modification to decomposition functions

As part of the development of this version of JULES-Peat we improved the response of soil carbon decomposition both to soil
moisture and nitrogen availability. These changes were made based on well-known principles. Firstly, that microbial activity
drops to zero in completely dry conditions (Yan et al., 2018); secondly that respiration in anaerobic conditions is known to
be no higher than 20% of the maximum rate in aerobic conditions (Schuur et al., 2015); and finally that when microbes lack
nitrogen, they tend to decompose plant litter *faster* in order to 'mine' for nitrogen (Craine et al., 2007) in contrast to the original
scheme introduced by Wiltshire et al. (2021) in which the decomposition of litter is inhibited when nitrogen is in short supply.

The decomposition of soil carbon in JULES is calculated as follows: For each soil carbon pool ($C_p$ kgm$^{-2}$, where $p$ denotes
the pool number), the potential turnover - i.e. the turnover rate when the nitrogen in the system is not limiting - is given by
($R_{p,pot}$):

$$R_{p,pot} = k_p C_p F_T(T_{soil}) F_\theta(\theta) F_v(v) \tag{1}$$

where $k_p$ is a fixed constant for each pool in s$^{-1}$ (Clark et al., 2011). The functions of temperature ($F_T(T_{soil})$) and moisture
($F_\theta(\theta)$) depend on the temperature ($T_{soil}$, K) and moisture content ($\theta$, fraction of saturation) of the soil. The function $F_v(v)$
depends on the vegetation cover fraction ($v$) (Clark et al., 2011). When the vertically-resolved soil carbon scheme is used,
there is an additional multiplier, $\exp(-z\tau_{resp})$, where $z$ is depth in the soil and $\tau_{resp}$ represents an additional decay of carbon
decomposition rate with depth.

$F_\theta$ is a function of the soil moisture. The standard version of JULES uses the following function which is also shown in
green on Figure 1F:

$$F_\theta(\theta) = \begin{cases} 1 - 0.8(\theta - \theta_o) & \text{for } \theta > \theta_o \\ 0.2 + 0.8\left(\frac{\theta - \theta_{min}}{\theta_o - \theta_{min}}\right) & \text{for } \theta_{min} < \theta \le \theta_o \\ 0.2 & \text{for } \theta \le \theta_{min} \end{cases} \tag{2}$$

where $\theta_o = 0.5(1 + \theta_w)$, where $\theta_w$ is the wilting point water content as a fraction of saturation, and $\theta_{min} = 1.7\theta_w$. $F_\theta$ takes a
value between around 0.6–0.85 in saturated conditions, i.e. decomposition rate in saturated conditions is between 60–85% of
its maximum rate. However, in reality, aerobic respiration essentially stops in saturated conditions, and anaerobic respiration
takes place instead, with a rate less than 20% of the maximum aerobic respiration rate (Schuur et al., 2015). The fact that
decomposition is suppressed under saturated conditions is key to the formation of peat. Therefore, we modified the decompo-
sition function so that it takes a value of 0.2 when the soil moisture is saturated. We also changed the behaviour of this function
under dry conditions, since there are a number of studies available that indicate the shape of this function (Moyano et al., 2012,





2013; Yan et al., 2018), which should increase in a close-to-linear manner from zero decomposition rate at zero soil moisture
content.

In addition, for undecomposed organic soils specifically, critical and wilting point soil moisture can be very small due to
the large pore spaces and thus low capillary suction (critical point can be as low as 10% saturation). The formulation of $\theta_o$,
on the other hand, limits the optimum soil moisture content for respiration to a minimum of 50% saturation, which can be up
$5\times$ higher than the critical point. The critical point is defined by a capillary suction of 3.36 m. Moyano et al. (2013) show on
their Figure 3b that the respiration response to soil moisture reaches a maximum at around this value. They show a moisture
response curve for a high carbon soil in their Figure 3a. The curve reaches a maximum at around 30-40% saturation and stays
at a high value until ~75% saturation, in contrast to the formulation in JULES which reaches its maximum only at the point $\theta_o$.
Therefore, for soil layers in which the critical soil moisture is lower than $\theta_o$, we set the soil respiration to reach its maximum
at the critical soil moisture content, and remain at its maximum value until the original maximum $\theta_o$, resembling the 'high
C content' curve in Moyano et al. (2013) Figure 3a. We therefore define a 'lower' $\theta_o$, $\theta_{o,l} = \min(\theta_{crit}, \theta_o)$. The old and new
functions are shown in Figure 1F.

The new function in JULES-Peat is therefore:

$$F_\theta(\theta) = \begin{cases} 1 - 0.8 \left( \frac{\theta - \theta_o}{1 - \theta_o} \right) & \text{for } \theta > \theta_o \\ 1 & \text{for } \theta_{o,l} < \theta \leq \theta_o \\ \frac{\theta}{\theta_{o,l}} & \text{for } 0 \leq \theta \leq \theta_{o,l} \end{cases} \tag{3}$$

In the standard version of JULES, in situations where nitrogen is limiting, the decomposition of the litter carbon pools
(Decomposable Plant Material; DPM, and Resistant Plant Material; RPM) is reduced. This is because the more decomposed
pools have a higher nitrogen content - or lower C:N ratio - and therefore to decompose the litter carbon into the BIO (Biomass)
and HUM (Hummus) pools requires a source of nitrogen - nitrogen is 'immobilised'. Thus the decomposition terms $R_{pot}$ for
DPM and RPM pools are multiplied by a factor $F_N$, which is given by:

$$F_N = \frac{(M_{BIO} + M_{HUM} - I_{BIO} - I_{HUM})\Delta t + N_{in}}{(D_{DPM} + D_{RPM})\Delta t} \tag{4}$$

where $N_{in}$ is the total soil inorganic N pool in kg [N] m$^{-2}$, $M_p$ and $I_p$ are mineralisation and immobilisation of nitrogen
respectively, from pool $p$ in kg [N] m$^{-2}$ s$^{-1}$ and $\Delta t$ is the time step. $D_{DPM}$ and $D_{RPM}$ are the net demand associated with
decomposition of each of the litter pools:

$$D_p = I_{p,pot} - M_{p,pot} \tag{5}$$

where here $p$ is one of $RPM$ or $DPM$. See Wiltshire et al. (2021) for details.

However, in reality the microbes would continue decomposing the litter pools in order to access the nitrogen for their own
survival. They would not be able to transform all of the decomposed carbon into biomass due to lack of nitrogen, but the carbon
would decompose and would simply be released to the atmosphere as carbon dioxide, i.e. their carbon use efficiency reduces
(Manzoni et al., 2012). Therefore, instead of modifying the litter decomposition rate with the factor $F_N$, we modify the fraction





of decomposed carbon that is released to the atmosphere vs stored in the soil. This means that the limitation term has to take a

slightly different form. The new function is:

$$F_N = \frac{(M_{BIO} + M_{HUM} - I_{BIO} - I_{HUM} + M_{DPM} + M_{RPM})\Delta t + N_{in}}{(I_{DPM} + I_{RPM})\Delta t} \tag{6}$$

The fraction of decomposed carbon that stays in the soil (rather than being released to the atmosphere), in other words the carbon use efficiency, is then multiplied by $F_N$ for the DPM and RPM pools. While nitrogen was not a focus of this study, the need for this modification became apparent once the soil column was allowed to expand with addition of plant litter. This led

to an unrealistic positive feedback in which litter carbon wasn't decomposed due to lack of nitrogen availability, meaning that as litter was added to the layer it took up an ever-large volume, eventually pushing the higher-nitrogen-containing pools out of the layer completely (further down the column) resulting in zero nitrogen availability, and forming unrealistically thick litter layers with no turnover.

### 2.3  Change of soil column height

Chadburn et al. (2017) showed that the typical soil profile simulated by ESM land surface schemes with vertically-resolved soil carbon (JULES and ORCHIDEE) displays a smooth decline with depth which resembles a mineral soil profile. However, in highly organic soils the soil carbon concentration typically increases with depth to a certain point before beginning to decline (Harden et al., 2012). This is because the density of the organic material in the surface is usually lower than in the deeper soil, so there is simply less material altogether in the surface layers, and therefore less carbon. The organic material in deeper layers

has a higher density: in part because it becomes compressed by soil/water above it, and in a large part because it is generally more decomposed.

The crucial missing factor in global models (e.g. JULES, ORCHIDEE, CLM) is that the models don't account for the *volume* that is added to the soil when organic material is added via plant litter, or, conversely, the reduction in volume when organic material decomposes. This means that as well as being unable to simulate the typical profile of a peatland (soil carbon

increasing with depth near the surface), unrealistically high carbon contents in surface layers are often simulated. (See original JULES version on Figure 4, red lines).

In JULES-Peat, the profile of litter inputs into each soil layer and decomposition of soil carbon in the layer is calculated as previously (Burke et al., 2017a), except that the modified decomposition function is used (Section 2.2). When these increments come to be applied to the soil carbon profile, however, the thickness of the soil layers is now recalculated based on the volume

of organic matter added/removed. We calculate the change in layer thickness by prescribing a density to each carbon pool, using a higher density for the decomposed carbon pools than the litter carbon pools. After addition/removal of carbon in a given timestep, the new effective thickness $dz_{eff,n}$ of soil layer $n$, relative to the initial layer thickness $dz_n$ is given by:

$$dz_{eff,n} = dz_n + (dC_{\mathrm{dpm},n} + dC_{\mathrm{rpm},n})/(f_c\rho_{\mathrm{dpmrpm}}) + (dC_{\mathrm{bio},n} + dC_{\mathrm{hum},n}))/(f_c\rho_{\mathrm{biohum}}) \tag{7}$$

where the $\rho$ are the bulk densities associated with the carbon pools in $\mathrm{kg\,m^{-3}}$, $f_c$ is the fraction of organic material that is

carbon and $dC_{i,n}$ is the increment in carbon pool $i$ in soil layer $n$. We picked the density of the litter pools, $\rho_{dpmrpm}$ to be





the lowest density that is typically measured for peat (where DPM and RPM are the two litter carbon pools in JULES), and the density of the more decomposed carbon pools $\rho_{biohum}$ to be the highest density that is typically observed for peat (where BIO and HUM are the more decomposed carbon pools in JULES). Thus the bulk density of organic material in any given soil layer will fall somewhere between these two extreme values given that each layer typically contains all 4 carbon pools albeit

in different ratios. The values we chose were 35 kgm$^{-3}$ as the minimum, $\rho_{dpmrpm}$, and 210 kgm$^{-3}$ as the maximum, $\rho_{biohum}$. These match well with commonly quoted literature values (e.g. Chambers et al. (2010)), and were derived from the 5th and 95th percentiles of the bulk densities in the global peat core dataset described in Section 3. These limits are shown by vertical dashed lines in Figure 1A-E. The maximum bulk density of 210 kgm$^{-3}$ corresponds well to the threshold bulk density for peat functioning in e.g. Wang et al. (2021). We relate the bulk density of the organic material to the carbon content by assuming that

$f_c = 0.56$, or in other words 56% of the organic matter is carbon, which was also based on the 95th percentile of the percent carbon in the peat core dataset from Gallego-Sala et al. (2018) (see Section 3) and is consistent with the range of observations, e.g. in Chambers et al. (2010).

The new layer thicknesses are labelled as "effective" layer thicknesses ($dz_{eff}$). In order to avoid technical difficulties and potential numerical problems with variable soil layer thickness (e.g. if surface layers become very thick), the soil carbon profile

is then interpolated back onto the original soil layers.

In order to interpolate the carbon profile, the carbon quantities ($C + dC$; kgm$^{-2}$) are first transformed to carbon densities $Cden$ in kgm$^{-3}$ by dividing them by the layer thicknesses, $dz$.

Then, the interpolation of the effective carbon density on the effective layers back into the original layers depends on whether the centre of the original layer is above or below the centre of the effective layer.

$$Cden_{n,i} = Cden_{eff,n,i} + \frac{3}{4}(Cden_{eff,n+1,i} - Cden_{eff,n,i})\frac{z_{eff,n} - z_n}{z_{eff,n} - z_{eff,n+1}} + \frac{1}{4}(Cden_{eff,n-1,i} - Cden_{eff,n,i})\frac{z_{eff,n} - z_n}{z_{eff,n} - z_{eff,n-1}}$$

195                                                                                                                                            (8)

if the original soil layer depth $z_n$ is deeper than the effective soil layer depth $z_{eff,n}$, and

$$Cden_{n,i} = Cden_{eff,n,i} + \frac{3}{4}(Cden_{eff,n-1,i} - Cden_{eff,n,i})\frac{z_{eff,n} - z_n}{z_{eff,n} - z_{eff,n-1}} + \frac{1}{4}(Cden_{eff,n+1,i} - Cden_{eff,n,i})\frac{z_{eff,n} - z_n}{z_{eff,n} - z_{eff,n+1}}$$

                                                                                                                                             (9)

if the original soil layer depth $z_n$ is shallower than the effective soil layer depth $z_{eff,n}$, where $z$ is the centre of each soil layer and $i$ indicates the carbon pool (DPM, RPM, BIO or HUM).

Mathematically, this represents an approximated second order Taylor expansion of the function $Cdens(z)$ around the point $z_{eff}$ but with a particular choice regarding the second order derivative. In order to preserve the vertical structure of the soil, the second order derivative is assumed to be around $z + \delta z$ and $z - \delta z$, so if there is a 'corner' in the function it will not be smoothed out. This means that a peat layer will not end up being numerically smeared into the rest of the profile. This is explained in detail with equations in the supplementary material. Briefly, we used a simple test model where soil carbon inputs and outputs





are given prescribed input and turnover rates, we account for the expansion and contraction of the soil column when carbon
is added or removed, and tested the method of interpolating back onto the original soil layers (i.e. as used in JULES). We ran
this simple (and thus much quicker to run) script with very high resolution soil layers to see what the 'true' solution for the
soil carbon profile would be. We therefore confirmed that our choice of second order derivative gave the best approximation
of the true solution when a lower-resolution soil was used. For details, see Supplementary material. Supplementary Figure S2

shows that in the chosen scheme there does still appear to be some 'smearing' in the deeper layers, which are thicker, but using
a smaller interval $\delta z'$ leads to numerical instability in the thinner, surface layers. Thus, for future development a scheme where
$\delta z'$ depends on the soil layer thickness could be considered.

While the carbon profile is interpolated onto the original soil layers in order to keep the layer thicknesses constant, the
thickness of the deepest soil layer is updated in order to track the overall change of soil column height. The minimum thickness

for that layer is taken from the soil layer thicknesses specified at runtime, and this corresponds to the layer thickness when
there is no carbon in the soil. This base layer is extended based on the total volume of the carbon pools in the soil column,
and this extension is considered to be the surface elevation. However, the extra thickness of the bottom layer is neglected
when calculating fluxes of heat and water, and only applies when calculating carbon and nitrogen stocks and fluxes (which are
conserved during layer adjustments). We considered the complexity of modifying the heat and water calculations to account

for the variable-thickness base layer was not worth the added complexity, given that the fluxes at the base of the soil column
(8m depth in this case) are generally very small.

### 2.4    Simulating the age profile

Peat age was simulated following a similar method to Burke et al. (2017a) where the fraction of old carbon is traced throughout
the simulation. During each update of the soil carbon pools, the age of each carbon pool is tracked and the weighted average

of the soil age is taken for each carbon pool in each soil layer.

Each soil carbon pool in each soil layer is assigned an age, $A$, at the start of the simulation, which currently is either zero on
initialising the spin-up or is initialised from a dump file. Each time the carbon pools are updated, the age of each soil carbon
pool in each layer is increased by the timestep length. These values are then modified as the soil carbon pools are updated,
either due to input of fresh carbon from litter (which has an age of zero and therefore reduces the age of the soil pool), due to

mixing of carbon between two layers in which the ages are different, or due to input of carbon into BIO and HUM pools from
other pools via decomposition. The general formula to update the age ($A$) for carbon pool $C_i$ (kg m$^{-2}$), with an increment of
carbon $C_i \to C_i + \Delta C_i$ is:

$$A_{C_i} \to \frac{A_{C_i} C_i + A_{\Delta C_i} \Delta C_i}{C_i + \Delta C_i} \tag{10}$$

$\Delta C_i$ includes both incoming and outgoing fluxes from the pool. For the outgoing fluxes in $\Delta C_i$, we assume that $A$ is the

same as for the $C_i$ pool. For an incoming litter flux we assume that $A$ is zero, and incoming fluxes from other pools naturally
take the age value from the corresponding pool.





If the soil height accumulation is switched on, the age then must be interpolated back onto the original soil layers as described in Section 2.3. We use the same interpolation method as for the soil carbon, given in Equations 8 and 9.

### 2.5 Coupling between properties and C concentration

In order to dynamically update the soil physical characteristics, we assume that the physical properties of the organic material in the soil are a function of its bulk density. The bulk density that we simulate in JULES depends on how decomposed the soil carbon is. More highly decomposed organic matter has a higher bulk density, and its properties change as it decomposes. Notably, the hydraulic conductivity becomes much lower as bulk density (or decomposition) increases, which is included in other peat models (Frolking et al., 2010; Young et al., 2017), but we also fitted relationships between the bulk density

and other key physical characteristics, namely porosity, saturated hydraulic suction, Clapp-Hornberger exponent and thermal conductivity. For the heat capacity we assumed that the heat capacity of the organic material does not significantly change with decomposition status, and therefore we used $(1-$porosity$)$ (i.e. what fraction of the organic material is solids) multiplied by the heat capacity value of solids of $2.5 \times 10^6$ JK$^{-1}$m$^{-3}$ which we took from Beringer et al. (2001).

Since different carbon pools have different bulk densities (see Section 2.3), we first calculate the bulk density of the combined

organic material in each soil layer, i.e.

$$\rho_{org,n} = \frac{1}{f_c} \sum_i C_{n,i}/(f_{org,n}dz_n) \tag{11}$$

where $f_{org,n}$ is the volumetric fraction of organic matter in the soil layer, given by:

$$f_{org,n} = ((C_{n,1} + C_{n,2})/\rho_{dpmrpm} + (C_{n,3} + C_{n,4})/\rho_{biohum})/dz_n \tag{12}$$

Recent studies have shown that bulk density of peat shows strong relationships with its thermal and hydraulic properties

(Liu and Lennartz, 2019; O'Connor et al., 2020). We combined data from these recent syntheses with additional values from literature in order to get the best estimate of the relationships, which we show in Figure 1. We fitted the relationships between bulk density and the other physical characteristics of peat using this combined dataset. Fitting was done using orthogonal least squares after normalising the data so that both variables being fitted had the same range of values. For the saturated hydraulic conductivity the two available datasets showed markedly different relationships (see Figure 1B), and so we did not combine

these but instead used only the data from Liu and Lennartz (2019) since this was firstly a global synthesis as opposed to the O'Connor et al. (2020) data which was from a single region. The data in Liu and Lennartz (2019) also agreed better with other data such as Wang et al. (2021), and the original values used for organic soils in JULES, originally given in Dankers et al. (2011) (also shown on Figure 1). In addition, the fit for porosity was forced to pass through 1 at a bulk density of zero, as a physical constraint (this was achieved by modifying the normalisation factor until the intercept of the fit was exactly 1).

The additional literature data for saturated hydraulic suction and Clapp-Hornberger exponent shown in purple on Figure 1 were derived from the following papers: Londra (2010); Rezanezhad et al. (2012); Da Silva et al. (1993); Weiss et al. (1998); Päivänen et al. (1973); Boelter (1964); Rydén et al. (1980); Schwärzel et al. (2006).



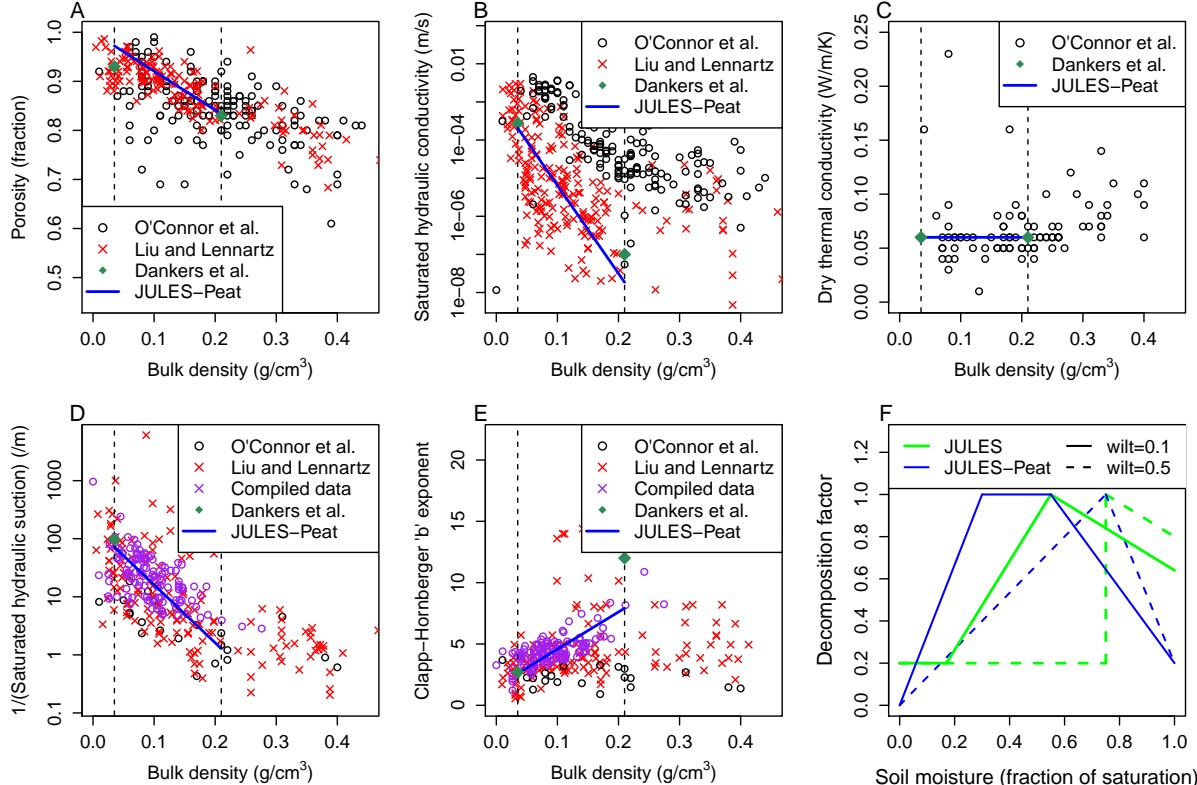

**Figure 1.** Functions used in JULES-Peat.

Specifically, we relate the following soil properties to bulk density:

$$\Psi_{sat,org} = \exp\left(0.023\rho_{org} - 5.08\right) \tag{13}$$


$$b_{org} = 0.0304\rho_{org} + 1.53 \tag{14}$$

$$K_{sat,org} = \exp\left(-0.0532\rho_{org} - 6.63\right) \tag{15}$$

$$\theta_{sat,org} = 1 - \rho_{org}/1260 \tag{16}$$

$$\lambda_{org} = 0.06 \tag{17}$$





$$hcap_{org} = 2.5 \times 10^6 \rho_{org}/1260 \qquad (18)$$

where $\Psi_{sat,org}$ is soil matric suction at saturation (m), $b_{org}$ is the Clapp-Hornberger exponent (unitless), $K_{sat,org}$ is hydraulic conductivity in units of $\mathrm{ms}^{-1}$ (note that JULES uses $\mathrm{kgm}^{-2}\mathrm{s}^{-1}$ so this is multiplied by 1000 for use in JULES), $\theta_{sat,org}$ is the volumetric soil moisture at saturation ($\mathrm{m}^3\mathrm{m}^{-3}$), $\lambda_{org}$ is the dry thermal conductivity ($\mathrm{Wm}^{-1}\mathrm{K}^{-1}$) and $hcap_{org}$ is the dry heat capacity ($\mathrm{Jm}^{-3}\mathrm{K}^{-1}$). If a soil layer is not 100% organic then we combine these calculated organic parameters with the properties of the underlying mineral soil, following Chadburn et al. (2015a). The remaining hydraulic parameters $\theta_{wilt}$ and

$\theta_{crit}$, which are the volumetric soil moisture at the wilting point and critical point respectively (defined in terms of hydraulic suction) are functions of the other parameters and are recalculated when the other parameters are updated (see e.g. Chadburn et al. (2015a)).

## 3 Simulations and evaluation data

We used a large suite of simulations at 24 sites that have been use for JULES development and evaluation in Chadburn et al.
(2017, 2020); Nakhavali et al. (2018); Gao et al., in prep.; Smith et al., in prep., along with Scotty Creek (Helbig et al., 2016, 2017a,b), Pleistocene Park (Euskirchen et al., 2017b), Imnavait (Euskirchen et al., 2017a), and Eight Mile Lake (Celis et al., 2019). The sites are fairly evenly distributed between tundra, boreal and temperate climate zones - see Table 1. Some of the sites, namely Abisko, Seida and Imnavait, are split into different landscape types, resulting in 29 simulations in total. The climate forcing data was prepared as described in Chadburn et al. (2017) using WFD and WFDEI (Weedon et al., 2011, 2014)
corrected with local climate data from the sites, and covers the period 1901-2018 inclusive. The simulations were spun up for 10,000 years using repeated climate forcing data from 1901-1910.

The initial JULES simulation (JULES, Table 2) is based on the configuration in Chadburn et al. (2020), but now has 20 soil layers extending to around 7.9 m, with thicknesses given in Supplementary Table S3. This was originally derived from the JULES-ES configuration (see https://jules.jchmr.org/content/core-configurations), with extra processes added to enhance the
simulation, particularly for high latitudes. For example, an extra PFT is included to represent Arctic grass, based on C3 grass with temperature optimum adjusted to grow in colder climates; layered soil carbon and nitrogen are switched on; a bedrock column is included below the soil to simulate heat conduction. Starting from this baseline simulation, we then switched on the new processes in JULES-Peat, described in Section 2. See Code and Data Availability, below, for the full configuration.

For most of the simulations the standard TopModel-based large-scale hydrology scheme was used, which calculates the
lateral flow of water from each grid cell based on the topographic index information of the grid cell (Gedney and Cox, 2003). In order to simulate a wetter site, for example a topographically controlled peatland which would essentially be a wetter fraction of the grid-cell than the grid-cell average, we simply set the lateral flow to zero (JULES-Peat-W and JULES-Peat-W10, Table 2). Neither of these hydrological scenarios are necessarily expected to be realistic for the sites. The aim was to test the response of the model to wetter vs drier conditions. How to simulate peatland hydrology realistically is a challenge and will be addressed
in future work (see also Bechtold et al. (2019)).




**Table 1.** Sites used in the suite of JULES simulations. References are both for site data and for simulations of these sites with JULES.

| Site (simulation name) | Location | Climate zone | References |
|---|---|---|---|
| Abisko (Abisko; Abiskomire) | Sweden | Boreal | Jammet et al. (2017); Chadburn et al. (2017) |
| Abisko (Abiskomire_noSnowCor) | Sweden | Boreal | Jammet et al. (2017); Chadburn et al. (2020) |
| Auchencorth | UK | Temperate | Drewer et al. (2010); Gao et al., in prep. |
| Brasschaat | Belgium | Temperate | Gielen et al. (2010, 2011); Nakhavali et al. (2018) |
| Alberta - Western Peatland (CA_WP1) | Canada | Temperate | Long et al. (2010); Flanagan and Syed (2011) Gao et al., in prep. |
| Carlow | Ireland | Temperate | Walmsley et al. (2011); Nakhavali et al. (2018) |
| Chersky | Russia | Tundra | Kittler et al. (2017); Göckede et al. (2019); Chadburn et al. (2020) |
| Degerö | Sweden | Boreal | Nilsson et al. (2008); Sagerfors et al. (2008); Gao et al., in prep. |
| Eight Mile Lake (EML) | USA (Alaska) | Boreal | Celis et al. (2019) |
| Hainich | Germany | Temperate | Kutsch et al. (2010); Schrumpf et al. (2011); Nakhavali et al. (2018) |
| Imnavait (ImnavaitRidge, ImnavaitTussock and ImnavaitFen) | USA (Alaska) | Tundra | Euskirchen et al. (2017a) |
| Iskoras | Norway | Tundra | Kjellman et al. (2018); Smith et al., in prep. |
| Kopytkowo | Poland | Temperate | Fortuniak et al. (2021); Gao et al., in prep. |
| Kytalyk | Russia | Tundra | Van der Molen et al. (2007); Parmentier et al. (2011) Chadburn et al. (2017) |
| Lompolojänkkä | Finland | Boreal | Aurela et al. (2009); Lohila et al. (2010); Chadburn et al. (2020) |
| Mer Bleue Bog (Merbleue) | Canada | Boreal | Moore et al. (2011); Brown et al. (2014); Gao et al., in prep. |
| Pleistocene Park (PleistocenePark) | Russia | Tundra | Euskirchen et al. (2017b) |
| Samoylov | Russia | Tundra | Boike et al. (2019); Chadburn et al. (2015a) |
| Scotty Creek (Scottycreek) | Canada | Boreal | Helbig et al. (2016, 2017a,b) |
| Seida (Seidamin and Seidapeat) | Russia | Tundra | Marushchak et al. (2013); Biasi et al. (2014); Chadburn et al. (2020) |
| Siikaneva | Finland | Boreal | Zhang et al. (2020); Gao et al., in prep. |
| Svalbard Ny Alesund (Svalbard_Ny) | Norway (Svalbard) | Tundra | Boike et al. (2018); Chadburn et al. (2017) |
| Turkey Point (Turkeypt) | Canada | Temperate | Peichl and Arain (2006); Peichl et al. (2010); Nakhavali et al. (2018) |
| Twitchell | USA | Temperate | Valach et al. (2021); Miller et al. (2008); Miller and Fujii (2010); Chadburn et al. (2020) |
| Zackenberg | Greenland | Tundra | Elberling et al. (2008); Chadburn et al. (2017) |



**Table 2.** JULES simulations conducted. Note that T means 'True' (or the process is switched on) and F means 'False' (process switched off). F→T mean that the process was switched off during spinup and on during the main run. The 'Decomp. function' refers to changing from the original to the new decomposition function show in Figure 1F, and we also changed a switch that was using the total soil moisture instead of the unfrozen soil moisture, so that the new decomposition function is a function of the unfrozen soil moisture (more realistic, since frozen water is not available for microbes to use). Where the Initial C is given as 'Peat' we initialise all spinups with the spun-up profile for Auchencorth from JULES-Peat-W, with ~1.5m of peat, and otherwise initalise the model with zero soil carbon.

| Simulation name | Accumulation | Decomp. function | $\tau_{resp}$ | $\tau_{lit}$ | Dynamic soil | Lateral flow | Initial C |
|---|---|---|---|---|---|---|---|
| JULES | F | F | 1.2 | 5 | F | T | 0 |
| JULES-Peat | T | T | 2 | 5 | T | T | 0 |
| JULES-Peat10 | T | T | 2 | 10 | T | T | 0 |
| JULES-Peat-W | T | T | 2 | 5 | T | F | 0 |
| JULES-Peat-W10 | T | T | 2 | 10 | T | F | 0 |
| JULES-Peat-i | T | T | 2 | 5 | T | T | Peat |
| JULES-Peat-i10 | T | T | 2 | 10 | T | T | Peat |
| JULES-Peat-W-drain | T | T | 2 | 5 | T | F→T | 0 |
| JULES-Peat-W10-drain | T | T | 2 | 10 | T | F→T | 0 |

The list of simulations is shown in Table 2. The impact of switching off the lateral flow (JULES-Peat-W and JULES-Peat-W10) and initialising the sites with peat (JULES-Peat-i and JULES-Peat-i10) - instead of letting the carbon build up from zero - was tested in JULES-Peat. Initialising with peat soil tests whether the model retains a peat layer at sites where peat was not able to form from scratch, which would indicate self-regulating functions of peat in JULES-Peat. In addition to these simulations,

new processes were firstly switched on one by one in factorial until the full 'JULES-Peat' was run - these simulations are shown in the Supplementary material. During this process a few different parameter combinations were tested to make sure the soil carbon profile and the age-depth profile looked realistic. In particular we altered the rate of decay of soil respiration with depth (efolding), $\tau_{resp}$, and the efolding depth of litter inputs to the soil, $\tau_{lit}$ (called $\xi_{lit}$ in Wiltshire et al. (2021)). A higher value of $\tau_{lit}$ means more of the litter added to the surface compared to deeper in the soil. We ran JULES-Peat with two

different values of $\tau_{lit}$, 5 and 10, where 5 is the original value in JULES and any simulation with a '10' on the end (Table 2), has $\tau_{lit} = 10$, or in other words more of the plant litter added to the surface layers. Note that we did not develop the vegetation module further here, and this should be addressed in future work (Section 4.5).

For evaluation we used a globally-distributed dataset of peat profiles (Gallego-Sala et al., 2018). We divided these data into major climate zones and selected only those zones that are covered by the JULES simulations: temperate, boreal and

tundra. This left 216 sites: 12 tundra, 127 boreal and 77 temperate. We compared these peat profiles against the site simulations divided into the same climate zones. We also used this dataset to derive the values for maximum and minimum bulk density ($\rho_{dpmrpm}$ and $\rho_{biohum}$, Section 2.3) and the fraction of organic matter that is carbon ($f_c$). For this calculation we required only the datapoints that were essentially organic material with minimal mineral content, so we removed any datapoints for which





the percent carbon (by mass) was less than 30%, leaving only data where the vast majority of the soil by volume is organic
material. This left over 24,500 datapoints.

Before comparing the peat core dataset against the JULES simulations, we only remove values where the percent carbon is
less than 15% as we take this as a common definition of organic soil (Science and Administration, 1975), which can include
some mineral material. We use the same definition to assess where JULES-Peat simulates a peat or a mineral soil, noting that
to estimate the percent carbon by mass in JULES, we assume the mineral fraction of the soil has a bulk density of $1500 \, \text{kgm}^{-3}$
(Hossain et al., 2015). We isolate the peat layers from the JULES simulations in order to evaluate comparable soil layers against
the observed peat core dataset, and select only sites where JULES simulates peat in at least the top 4 soil layers (i.e. to 39cm
depth, since this is the closest layer to the 40cm depth specified for defining organic soils (Science and Administration, 1975)).

For further evaluation data we used individual soil carbon profiles from other data sources, which are available for some of
the sites in Table 1 (references given in the table).

## 4  Results and discussion

### 4.1  Representation of mineral soils

In soils where the organic material is a relatively small fraction of the total soil, the original soil carbon scheme is able to
perform well, since the expansion of the soil column due to the addition of organic material will be relatively small. Figure 2
shows the model performance at mineral soil sites where measured soil carbon profiles were available. While the individual
sites are not well simulated, the general form of the profiles - resembling an exponential decline with depth - are recreated
reasonably well.

Figure 2 also shows two JULES-Peat simulations in light green and dark green (JULES-Peat and JULES-Peat10, Table 2).
As discussed in Section 3 these have different values of $\tau_{lit}$, which are both plausible. In general, the new model version is
also able to simulate a profile that resembles a mineral soil, despite forming peaty profiles at a few of the sites, especially
Hainich and Carlow (where the carbon is overestimated by all versions of JULES). Aggregated across all sites, the updated
model versions produce a profile with somewhat lower carbon density at the surface compared to standard JULES, and less of a
decline in carbon with depth (final panel of Figure 2). The lower carbon density at the surface matches better with observations
than the original JULES simulation. Overall we conclude that mineral soil carbon profiles can be adequately represented with
all model versions.

### 4.2  JULES-Peat evaluation

We initially introduced the additional processes and parameter changes that were incorporated in JULES-Peat into simulations
one by one to test each process, before running the full model. These simulations are shown in the Supplementary material
(Supplementary Section 2).



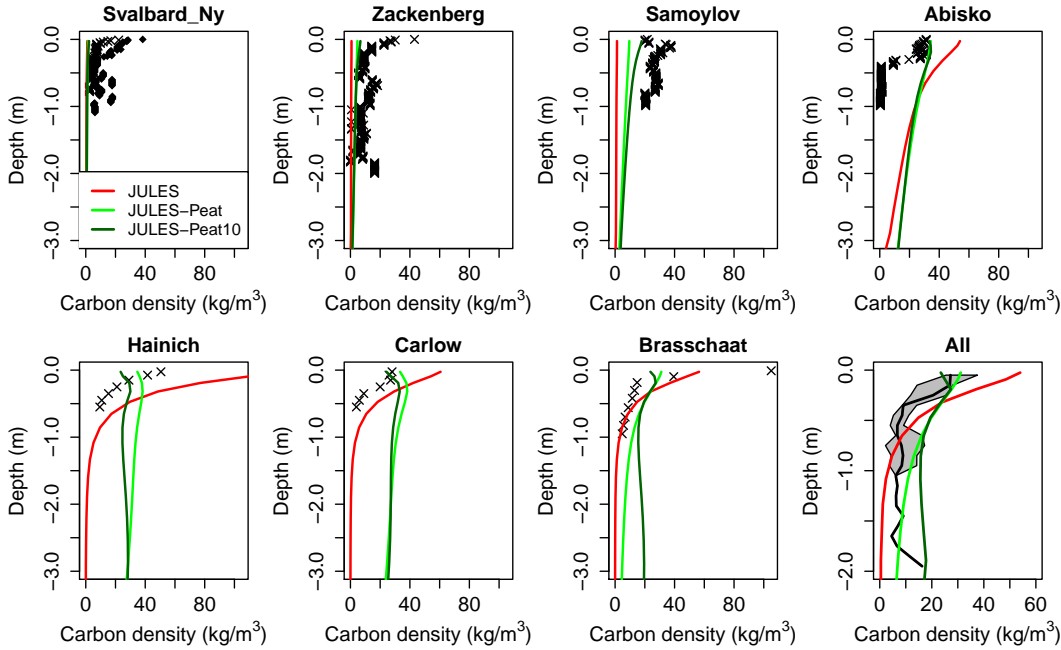

**Figure 2.** Evaluation at mineral sites. Note that the axis ranges are different on the final panel. For information about sites refer to Table 1 and for details of the JULES simulations see Table 2.

We assess the performance of the full JULES-Peat model at the selection of simulated sites for which observed soil carbon

profiles are available and organic soils are present (Figure 3). A few of the individual sites are well simulated, and almost all sites are simulated significantly better in all of the JULES-Peat configurations than they are in JULES (RMSE of median profile is given in Table 3). Note that two additional JULES-Peat simulations are shown in Figure 3, JULES-Peat-W and JULES-Peat-W10 (dark blue and light blue lines), where the lateral water flow is set to zero since we expect that this would lead to a wetter soil and a more realistic simulation of a topographically-controlled peatland.

We then evaluate the soil carbon and age-depth profiles in JULES and JULES-Peat against the global dataset of peat cores described in Section 3 (Gallego-Sala et al., 2018), Figures 4 and 5. These figures show simulated soil carbon profiles at sites where the model simulates a carbon percentage by mass of more than 15% for at least 39 cm (see Section 3), and compares these against the median of the equivalent data (percent carbon > 15%) from the global dataset of peat cores (Gallego-Sala et al., 2018). It is clear that at these peaty sites, the original JULES model simulates a carbon density that is too high in the

surface layers and too low in the deeper soil (red lines on Figure 4).

In order to quantify the total improvement in the various JULES-Peat simulations compared with the original JULES we take the root mean squared error (RMSE) between the median soil carbon profiles for each climate zone, shown in Table 3. The best-performing version (shown in bold) has a RMSE that is reduced by 35% for temperate peatland sites (from 23.0 kgm$^{-3}$ to 14.9 kgm$^{-3}$, Table 3) and by almost 80% for boreal peatland sites (RMSE reducing from 37.3 kgm$^{-3}$ to 7.7 kgm$^{-3}$, Table



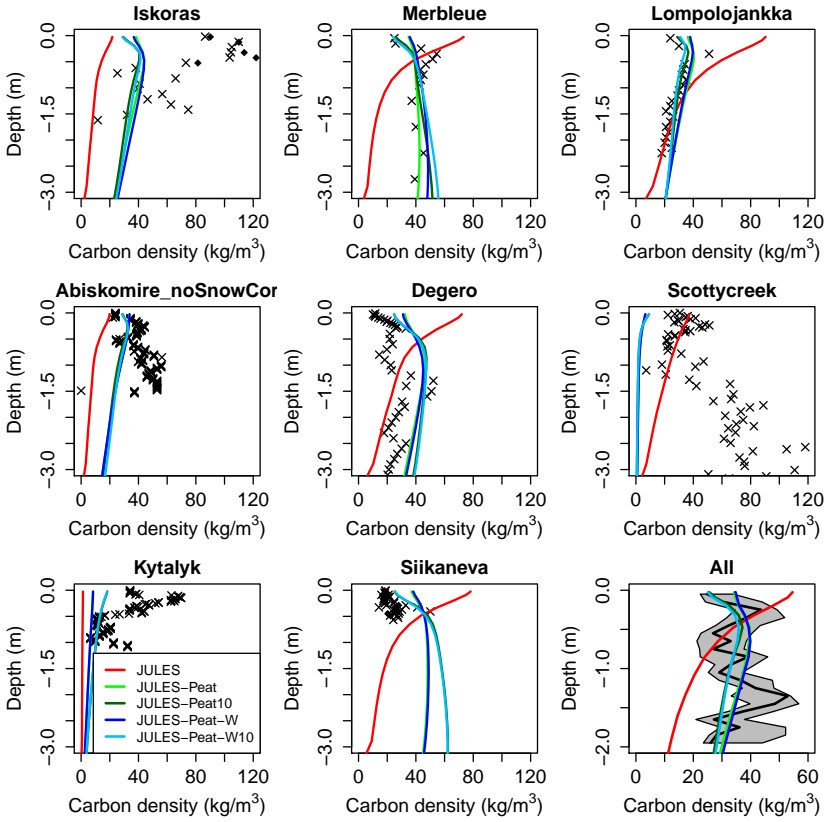

**Figure 3.** Evaluation at organic sites. Note that the axis ranges are different on the final panel. For information about sites refer to Table 1 and for details of the JULES simulations see Table 2.

3). We see that the age at the soil surface was typically too high in the original version of JULES (Figure 5, red lines). In JULES-Peat, the age at the soil surface is better simulated, and age throughout the profile is generally realistic (blue and green solid lines on Figure 5, mostly falling within the interquartile range of the observations), and RMSE in age is reduced by 32% and 56% for the configurations that perform best in terms of carbon profile (Table 3).

     The best simulations in JULES-Peat are generally those where drainage is impeded to make the soil wetter (JULES-Peat-W,
see bold numbers in Table 2). In particular, JULES-Peat-W simulates a more realistic carbon density profile than the other model configurations for the temperate peatland sites (see dark blue lines in Figure 4). For the boreal sites, JULES-Peat-W10 accurately captures the gradient of the soil carbon profile in the top 50cm of soil (light blue lines in Figure 4). The age-depth profiles in JULES-Peat-W and JULES-Peat-W10 also correspond most closely to the median measured age-depth profiles. Since peat generally forms in wetter places, the fact that the simulations without lateral water flow out of the soil (-W and
-W10) compare best against observations is an indication that the model behaviour is reasonable.



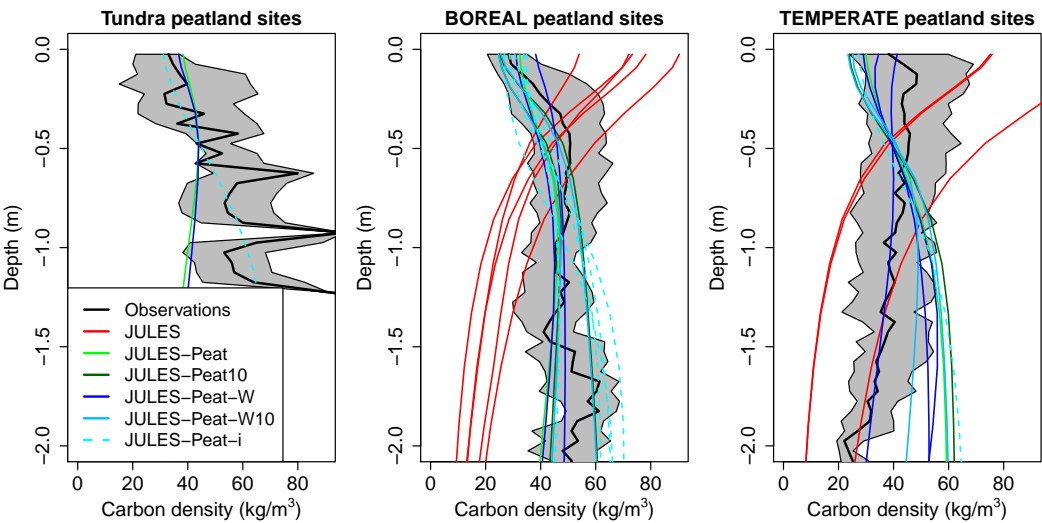

**Figure 4.** Evaluation of JULES-Peat carbon profile against typical profile for peat cores.

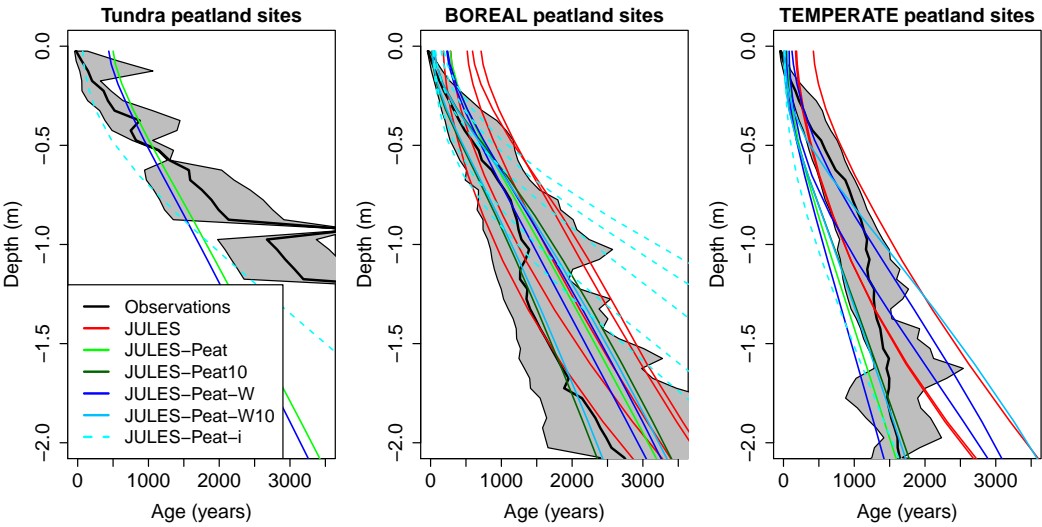

**Figure 5.** Evaluation of JULES-Peat age-depth profile against typical profile for peat cores.

In simulations whose name ends with '10' (e.g. JULES-Peat-W10, shown in light blue) the distribution of litter inputs into the soil is more weighted towards the surface ($\tau_{lit} = 10$ as opposed to 5, Table 2). We generally find that using the lower value of $\tau_{lit}$ matches better with the data for the temperate peatlands and the higher value is better for boreal peatlands (see RMSE values for carbon profile in Table 3). This suggests that $\tau_{lit}$ should depend on plant functional type, which it would in reality

(shallower-rooting plants would deposit more of their litter nearer the surface), and implies that smaller/shallower-rooting PFT's should grow in colder regions, as is indeed the case (Jackson et al., 1996).



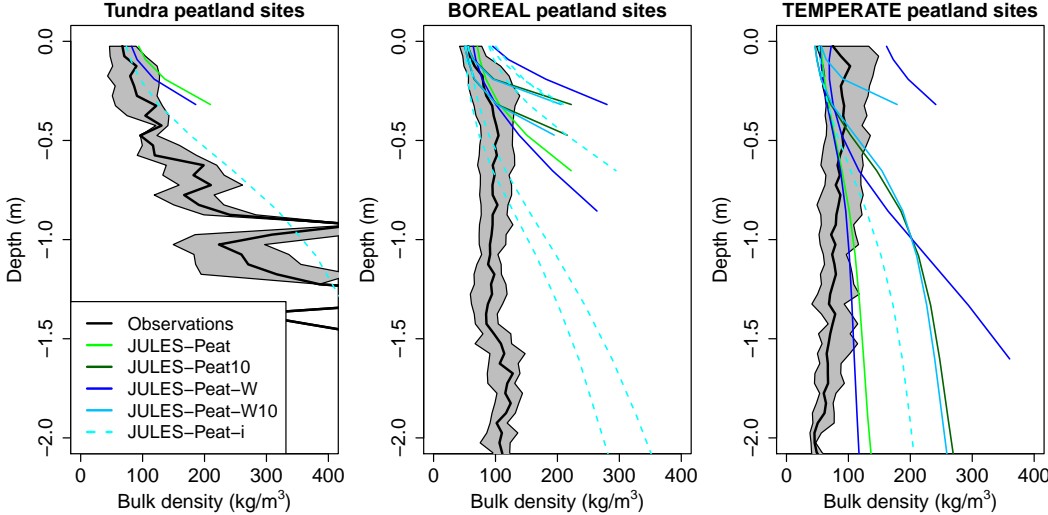

**Figure 6.** Evaluation of JULES-Peat bulk density profile against typical profile for peat cores. Profiles are cut off because only the organic layers are plotted.

JULES-Peat was only able to accumulate peat from scratch at one of the tundra sites - Iskoras - and only in selected configurations (JULES-Peat and JULES-Peat-W, see Supplementary Figure S12). These simulations form a relatively thin organic layer at this site. However, when the simulations were initiated with an existing peat profile instead of zero soil carbon, in JULES-Peat-i (see Table 2), then the peat profile was maintained at Iskoras and continued to accumulate (Supplementary Figure S12). This then forms a realistic carbon density profile and a reasonably realistic age-depth profile, shown by dashed cyan lines in Figures 4a and 5a. There are multiple reasons why the model may not accumulate much peat at tundra sites, including a lack of representation of more favourable paleoclimate conditions during spinup, and simulating soils that are too dry (Smith et al., in prep). Nonetheless, the Iskoras simulation by JULES-Peat-i indicates that when the model does simulate peat at a tundra site it can form a realistic profile (Figure 4a, dashed cyan line).

We also evaluated the bulk density profiles against the same peat core dataset, the results are shown in Figure 6. Again, the bulk density profile simulated for the tundra sites is realistic for the JULES-Peat-i simulation (dashed cyan line), where only the Iskoras site simulates a thick layer of peat. The bulk density in JULES-Peat tends to start at realistic values at the surface but to increase too quickly with depth. It is remarkable how little the observed bulk density at boreal and temperate peatland sites varies with depth compared with the tundra sites (Figure 6), although this may be related to the larger sample size of boreal and temperate sites (127 and 77, vs 12 for tundra) leading to a more 'smoothed' profile.

JULES-Peat simulates its own soil characteristics. While we don't have measured profiles of soil characteristics to compare against, we can compare the soil thermal and hydraulic parameters simulated by JULES-Peat against those prescribed in JULES. Comparisons are shown in the supplementary material for key parameters: $K_{sat}$, $\Psi_{sat}$, $\theta_{sat}$ and $b$, for the simulations JULES-Peat, JULES-Peat10 and JULES-Peat-i10 (Supplementary Figures S7–S10). For some sites there is a good correspon-





**Table 3.** RMSEs of JULES simulations against various observations. Temp't = Temperate. The final two columns refer to the median of the site-specific observations shown in Figures 2 and 3 respectively. The best performing simulation for each column is highlighted in bold.

| Simulation | C (whole profile, kgm$^{-3}$) | | | AGE (whole profile, years) | | | BD (peat layers only, kgm$^{-3}$) | | | Mineral Sites | Organic Sites |
|---|---|---|---|---|---|---|---|---|---|---|---|
| | Tundra | Boreal | Temp't | Tundra | Boreal | Temp't | Tundra | Boreal | Temp't | | |
| JULES | - | *37.3* | *23.0* | - | *1069* | *2872* | - | - | - | *12.1* | *25.5* |
| JULES-Peat | 31.9 | 16.0 | 22.6 | 1590 | 469 | **987** | 69.4 | 53.3 | 96.0 | **6.2** | 13.5 |
| JULES-Peat10 | - | **7.7** | 25.8 | - | 467 | 1081 | - | 59.0 | 179.4 | 8.1 | 13.4 |
| JULES-Peat-W | 30.8 | 12.2 | **14.9** | 1651 | **449** | 1954 | **53.0** | 83.2 | **87.7** | 13.8 | **12.9** |
| JULES-Peat-W10 | - | 7.8 | 17.9 | - | 461 | 1635 | - | **49.0** | 171.9 | 12.6 | 13.1 |
| JULES-Peat-i | **16.7** | 9.8 | 27.4 | **1212** | 2901 | 1194 | 69.6 | 157.8 | 133.5 | 28.5 | 19.7 |
| JULES-Peat-i10 | 28.0 | 13.7 | 30.4 | 4608 | 1861 | 1330 | 137.4 | 180.6 | 219.4 | 26.9 | 19.7 |

**Table 4.** Median water tables corresponding to the simulations shown in Figure 7

| Simulation | Colour in Fig. 7 | Water table depth (m) | | | |
|---|---|---|---|---|---|
| | | Auchencorth | Brasschaat | Carlow | Turkeypt |
| JULES-Peat | Green | 6.5 m | 19.5 m | 3.7 m | 25.0 m |
| JULES-Peat-W | Blue | 0.076 m | 0.33 m | 0.11 m | 0.55 m |
| JULES-Peat-W-drain | Purple | 6.9 m | 19.3 m | 4.7 m | 25.0 m |

dence between the simulated and prescribed parameters, and others there are significant differences, but all simulated profiles behave sensibly.

## 4.3 Drainage, subsidence and feedbacks between hydrology and soil carbon

For the simulations where the lateral flow of water was switched off during spinup in order to simulate a wetter and more
'peaty' soil (JULES-Peat-W and JULES-Peat-W10), we ran an alternative realisation of the 20th century where the lateral flow was switched back on (JULES-Peat-W-drain and JULES-Peat-W10-drain). At many sites the lateral flow is negligible in any case and this doesn't significantly affect the results. However, at some sites the 'wet' (-W) simulation maintains a water table near the surface whereas the standard simulation (JULES-Peat / JULES-Peat10) does not, and the drained simulations therefore experience a major drop in water table and a subsequent degradation of the peat and a drop in the surface elevation
(subsidence). Figure 7 shows the four sites for which the change in water table is most pronounced in JULES-Peat-W-drain (water table change given in Table 4).

Liu et al. (2020) tracked the surface subsidence rate over time following drainage in two different ecosystem types - forest and agriculture. They typically see a very high subsidence rate of around 3-10 cm per year in the first few years after drainage. Following that, a more steady subsidence rate of 0.5–2cm per year for the next few decades. In the sites on Figure 7 we see a



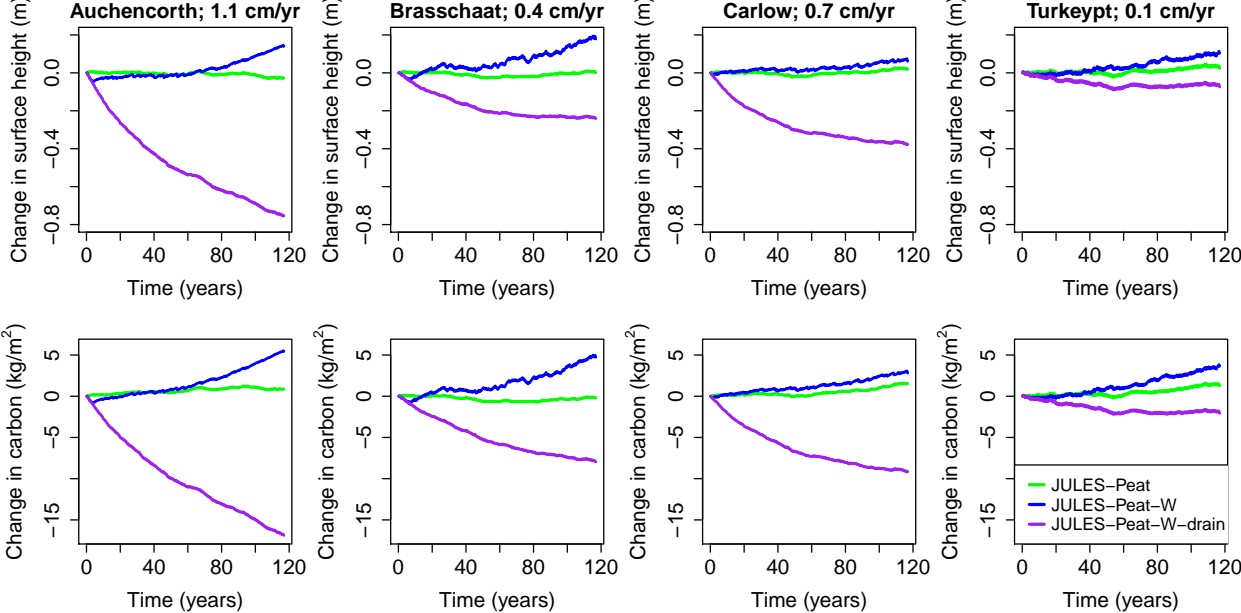

**Figure 7.** Carbon and surface height dynamics following drainage: JULES-Peat-W and JULES-Peat-W-drain are both spun up with a wet soil column, but JULES-Peat-W-drain has lateral flows switched back on at the start of the historical simulation, which is shown here. (JULES-Peat is also shown for comparison, showing that carbon does not accumulate at these sites to the extent that it does when the sites are wet, i.e. compare JULES-Peat to JULES-Peat-W). Subsidence rates in cm yr$^{-1}$ over first 40 years are indicated in the figure titles. Water table depth in each simulation is given in Table 4.

surface subsidence rate more in line with the longer-term subsidence rate of 0.5-2cm per year (e.g. Auchencorth loses >40cm in the first 40 years and 60cm in around 80 years, Figure 7). The lack of the initial very rapid subsidence suggests that there may be some processes missing in JULES, for example the mechanical raising and lowering of the peat surface as the water table fluctuates, known as bog breathing (Howie and Hebda, 2018). However at least the long term subsidence rate is the right order of magnitude. After an initial period of subsidence lasting around 50 years, the drop in surface height stabilises or slows.

The carbon loss behaves in a similar way, although the slowing of carbon loss is not as pronounced. In these test simulations (noting that the method of 'draining' the sites is a proof of concept and isn't based on reality), up to 17 kg C/m$^2$ is lost from these sites, which would represent a highly significant addition of carbon to the atmosphere - of the order of tens of Gt C globally - if it took place across a significant fraction of the world's peatlands.

Both positive and negative feedbacks exist within peatland ecosystems (Waddington et al., 2015). JULES-Peat is able to

capture some of the key feedbacks by simulating dynamic soil properties. Firstly, a negative [damping] feedback takes place following drainage in which the peat compacts and becomes more resistant to water flow, thus re-wetting the soil to some extent - see Figure 8. There is a strong correlation between decreased hydraulic conductivity and increased soil moisture (pearson correlation between soil moisture and $\log(K_{sat})$ of $-0.94$ for Auchencorth and $-0.87$ for Carlow, in the top 3m,



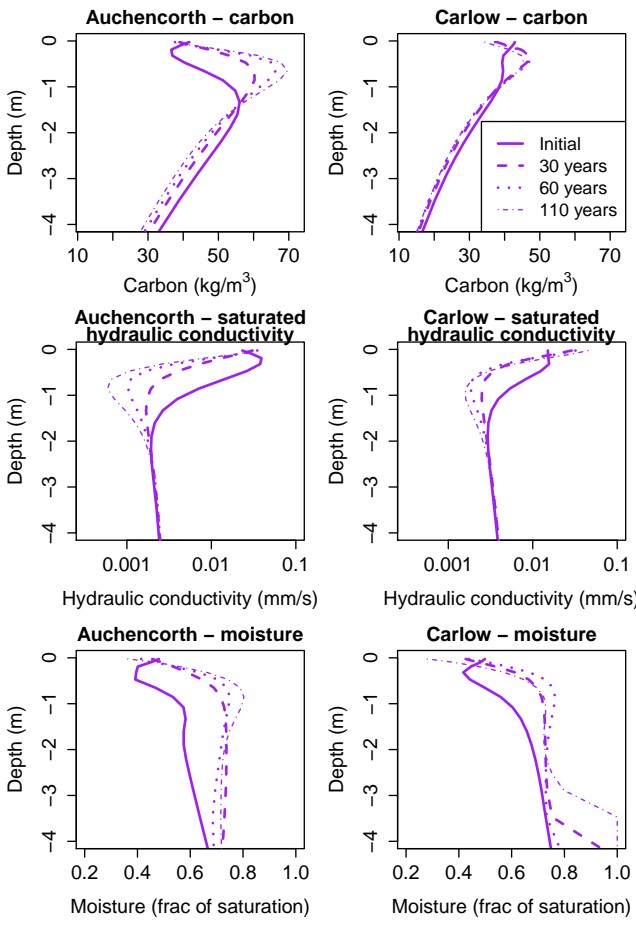

**Figure 8.** Response of the soil profile following drainage in JULES-Peat-W-drain.

using monthly data points for individual layers for the whole simulation). In these particular simulations, this effect was not

strong enough to bring the water table back to the surface by the end of the simulation, but in some test simulations this effect was observed. In reality, it is rare for drained peatlands to self-restore, but it does occasionally happen (Milner, pers. comm.; Angus, pers. comm.). On the other hand, a similar mechanism can lead to a positive [amplifying] feedback during spin-up, where the accumulation of peat leads to a lower drainage rate and thus further accumulation of peat. This is seen most strongly at Auchencorth and Carlow, which are the only sites from the UK and Ireland that were simulated (Figure 9). After sufficient

peat formation (in particular after being initialised with peat in JULES-Peat-i and JULES-Peat-i10, see Table 2), these sites are able to gradually raise their water tables over the course of the spin-up, while accumulating more and more carbon - see Figure 9. It is significant that the UK are Ireland sites are the only ones where this mechanism takes place, since this is where the majority of the world's blanket bogs are found - peatlands that are able to maintain themselves autogenically without topographic controls. This indicates that JULES-Peat should be capable of simulating blanket bogs, which are currently missing

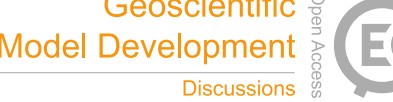

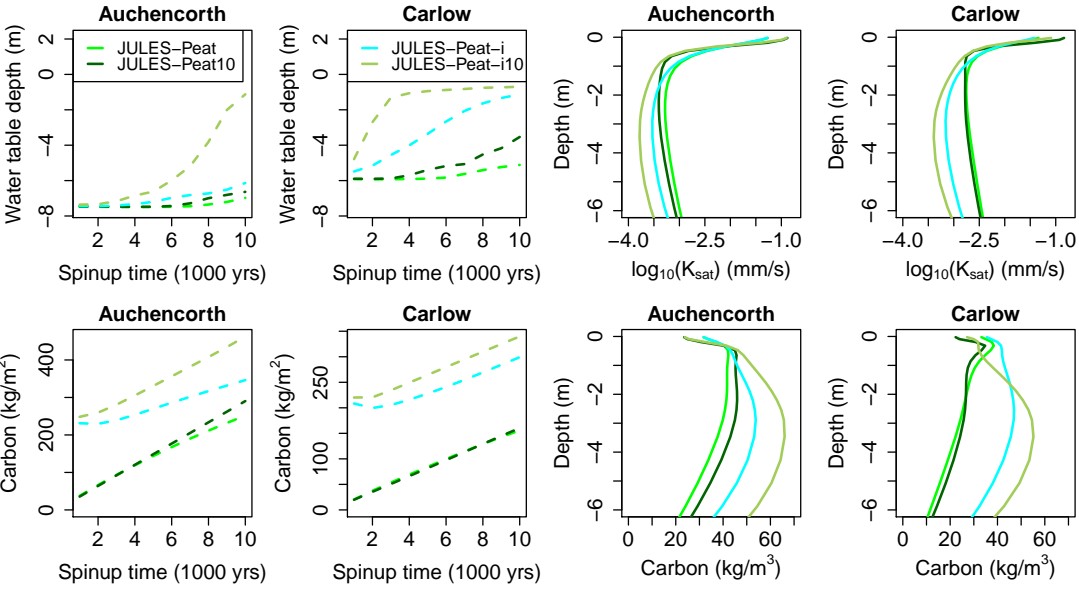

**Figure 9.** Hydrological feedback between peat accumulation and water table level. Note that the spinup time is noted in 1000's of years so the total spinup period is 10,000 years. Vertical profiles of $K_{sat}$ and carbon are shown at the end of the spin-up.

in global peatland models due to their reliance on TOPMODEL-based wetland fraction to determine peatland area (Müller and Joos, 2020).

### 4.4 Multiple steady states

Since there are feedbacks in JULES-Peat between soil physics and soil carbon that can be self reinforcing, this means that the 'end state' of the model spinup now depends on the initial conditions. In mathematical terms there can be more than one
equilibrium state. This also implies the existence of tipping points where the system can 'tip' from one state to another under sufficient forcing (Ritchie et al., 2021). Essentially, there may be some sites at which initialising the model with peat allows it to further accumulate peat, but initialising the model with mineral soil maintains a stable mineral soil profile. Practically, this means that peat can exist outside of climatic conditions where it would form from scratch, and so this is very important when we consider disturbance of existing peatlands for which the original state could be impossible to recover under current
climates.

In order to test this we compare the simulations where peat is initialised vs not initialised on Figure 10. We show two sites where peat only accumulates when it is initialised (Abiskomire_noSnowCor and Iskoras), a site where peat always accumulates (CA_WP1) and a site where peat never accumulates and a mineral soil always forms (Twitchell). This behaviour is apparent in the age-depth profiles (bottom row of Figure 10). The age is initialised with the existing age-depth profile of the peat for
the runs that are initialised with peat (JULES-Peat-i and JULES-Peat-i10), hence the ages overall can be higher, since the



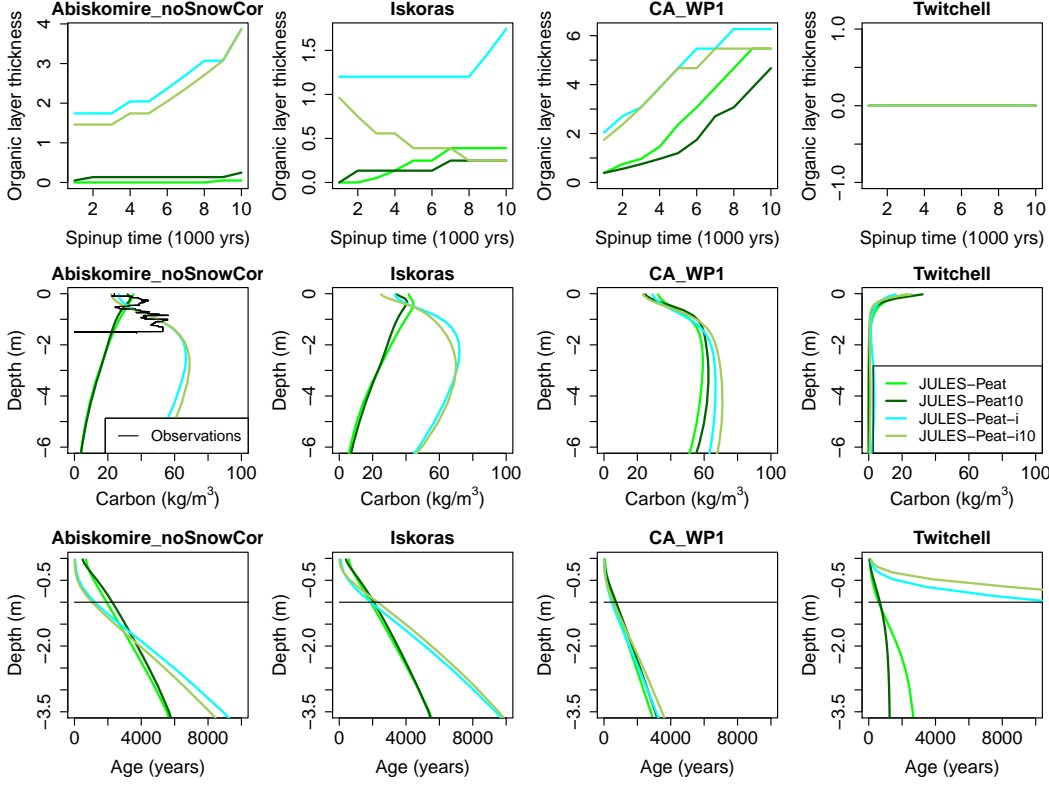

**Figure 10.** Demonstration of peat accumulating / not accumulating depending on initial conditions. Profiles of carbon and age are shown at the end of spin-up. Note that the ages below a certain depth at Twitchell are not relevant due to the carbon pools being almost zero. Note also that the spinup time is given in 1000's of years (so the total spinup period is 10,000 years).

standard runs start from age zero with no carbon. However, when the model converges to a single steady state, the age profile also starts to converge, at least at the surface (Twitchell and CA_WP1 on Figure 10). In contrast, at sites where peat only accumulates when the model is initialised with peat, the age at the surface actually becomes *lower* in the simulations where it was initialised with an existing age-depth profile than in the simulations where it was initialised at zero. This indicates that
carbon is accumulating more quickly when the model is initialised with peat (-i and -i10) (Abiskomire_noSnowCor and Iskoras on Figure 10).

It is interesting to note that the sites where highly distinct steady states are simulated are palsa mires in the sporadic permafrost zone. These sites are on the cusp of the permafrost/non-permafrost transition. Thawing of permafrost peatlands has been shown to increase the carbon accumulation rate in some cases (Turetsky et al., 2007), and, interestingly, the simula-
tions with high peat accumulation rate at Abiskomire_noSnowCor and Iskoras (JULES-Peat-i and JULES-Peat-i10) simulate a thawed soil, whereas the simulations with little peat accumulation (JULES-Peat and JULES-Peat10) simulate permafrost (Supplementary Figure S11).





**Table 5.** Dominant vegetation type and fraction at end of spin-up for the sites and runs shown on Figure 10. EG = Evergreen

| Simulation | Abiskomire_noSnowCor | Iskoras | CA_WP1 | Twitchell |
|---|---|---|---|---|
| JULES-Peat | Arctic grass, 0.59 | Arctic grass, 0.48 | Needleaf EG, 0.40 | C3 grass, 0.37 |
| JULES-Peat10 | Arctic grass, 0.60 | Arctic grass, 0.49 | Needleaf EG, 0.41 | C3 grass, 0.37 |
| JULES-Peat-i | Needleaf EG, 0.30 | C3 grass, 0.33 | Needleaf EG, 0.50 | C3 grass, 0.46 |
| JULES-Peat-i10 | Needleaf EG, 0.33 | C3 grass, 0.36 | Needleaf EG, 0.47 | C3 grass, 0.45 |

The different steady states also appear to be associated with the presence of different vegetation types, see Table 5. The sites that develop very different carbon profiles when initialised with peat (Abiskomire_noSnowCor and Iskoras, JULES-Peat-i and JULES-Peat-i10) also develop a different dominant vegetation type, for example needleaf evergreen trees instead of Arctic grass in Abiskomire_noSnowCor (Table 5). This interaction between vegetation and soil carbon highlights the importance of further developing the vegetation model to represent peatland vegetation (see Section 4.5 for further discussion). It is worth noting that for a site that simulates peat accumulation, the 'steady state' condition can be a constant rate of carbon accumulation rather than a constant quantity of carbon, i.e. peatlands are never in equilibrium, which differs from the standard definition of steady state that is currently used when spinning up land surface models.

### 4.5 Next steps for modelling global peatlands: Hydrology and vegetation

Simulating landscape-level peatland hydrology is a major challenge. This work has taken a step forwards by enabling the peat soils to react appropriately to long term changes in hydrology and to include some of the key feedbacks that the soils then have on the water flows. We have also recently developed an improved methane emissions scheme (Chadburn et al., 2020). However, to model peatlands within the landscape globally, including any methane emissions, the distribution of water around the landscape must be taken into account.

For instance, the majority of peatlands globally are topographically controlled. This means they are found in flat, lowland areas (Sheng et al., 2004; Martini, 2006). The standard way of modelling groundwater in ESMs does not explicitly model these areas, but simulates a 'grid cell average' soil moisture. This means at typical resolutions, the saturated lowland areas where peat forms would be less than the size of a grid cell and so saturated conditions would never be explicitly simulated, and secondly, even with a high enough resolution to resolve peatlands, there is no mechanism for lowland grid cell soils to receive water from the surrounding uplands. Therefore, a key step is to explicitly model different hydrological regimes/features within the landscape. The simplest way to do this is via a tiling approach. Bechtold et al. (2019) found that they were able to recreate the hydrological dynamics in the majority of the world's peatlands by using a tile that was entirely hydrologically disconnected from the rest of the grid cell. A further step would be to simulate the hydrological connection between the tiles in the grid cell, as this is not only necessary to simulate some existing peatlands but also to simulate peatland initiation (Väliranta et al., 2017).

Furthermore, the within-soil-column hydrology is not well modelled for organic soils in JULES. While peatlands in JULES-Peat are able to maintain a water table through the hydraulic characteristics of the peat itself, the water table generally does





not sit as close to the surface as the observed water table in intact peatlands, which is around 10 cm (Evans et al., 2021) (see
e.g. Carlow on Figure 9), although it can occasionally reach 10cm when lateral flow is set to zero, Table 4. In cold regions,
a representation of ponding can be necessary to prevent too much snowmelt from running off and leaving the soil too dry
(Smith et al., in prep). On top of this, the hydraulic behaviour of mosses, which form a primary component of high latitude and
temperate peatlands, is very different from that of vascular plants. Mosses do not extract water from the soil and it essentially
only evaporates from the surface, which could very well lead to a raised water table (Van Breemen, 1995). Thus the inclusion
of a moss PFT and its unique functions in land surface models like JULES should be a priority, and indeed several models have
done this (e.g. Porada et al. (2016); Chaudhary et al. (2017); Shi et al. (2021)).

It is clearly important to adequately represent the features of peatland vegetation. As well as the hydrological behaviour of
mosses, it will be crucial to include an appropriate distribution of plant litter inputs to the soil (see difference in the simulations
with different values of $\tau_{lit}$ on e.g. Figure 4), an appropriate recalcitrant litter fraction (for example mosses are more recalcitrant
than grass and therefore more likely to lead to peat accumulation), and suppression of the growth of non-wetland vegetation
such as trees under saturated conditions (this is not included in JULES and is necessary to simulate the mossy peat that is found
in northern latitudes, since larger vegetation would otherwise outcompete the mosses). In addition, the input of carbon to the
peat is determined by the net primary productivity of the ecosystem, and thus this is a key quantity to evaluate when developing
peatland-appropriate PFT's.

## 5 Conclusions and outlook

We have demonstrated a new scheme integrated in an ESM land surface model that can simulate both peat and mineral soils
depending on site conditions, and can simulate dynamic transitions from peat to mineral soil or vice versa. The model, which
we call JULES-Peat, includes some key ecohydrological feedbacks that take place in peat soils. At some sites, whether or not
peat accumulates depends on the initial conditions.

The model performs well by all metrics that we compared it against, and it can now simulate a soil profile that resembles
peat for the first time in JULES. As well as simulating mechanisms that determine the (in)stability and resilience of peatlands
for the first time, this model has the potential to simulate blanket bogs, which current global peatland models are unable to
do (Müller and Joos, 2020). We noted when designing the interpolation scheme (Supplementary information Section 1) that
the interpolation can lead to some 'smearing' of the carbon profile in the deeper soil, and indeed the model does not simulate
sharp transitions between peat layers and underlying mineral soil that can often be seen in reality. Thus some improvement to
the interpolation scheme may still be possible, which could also lead to improved physical soil characteristics. It should also
be noted that the JULES soil layers need to be set at a sufficiently high resolution to be capable of resolving such a transition.

As outlined above in Section 4.5, major challenges remain around appropriately modelling peatland vegetation and large-
scale hydrology. It may also be necessary to model microtopography and/or ponding in order to simulate soil hydrology
correctly (Smith et al., in prep). Since models individually tackle different parts of this problem, the next steps will inevitably
involve combining existing schemes, or at least concepts, for simulating vegetation, large-scale hydrology and microtopography



with the soil dynamics simulated here in JULES-Peat (e.g. Bechtold et al. (2019); Porada et al. (2016); Shi et al. (2021)), along with the latest methane emissions schemes (e.g. Chadburn et al. (2020)).

Peatlands are of utmost importance in terms of mitigating climate change, both as carbon sinks, but also as potentially very large carbon sources that may exacerbate climate change (Leifeld and Menichetti, 2018). Modelling global peatlands and their dynamics should therefore be a priority for land-surface and Earth system modelling.

*Code and data availability.*  Both the model code and the files for running it are available from the Met Office Science Repository Service: https://code.metoffice.gov.uk/. Registration is required and code is freely available subject to completion of a software license. The results presented in this paper were obtained from running JULES branch: https://code.metoffice.gov.uk/trac/jules/browser/main/branches/dev/sarahchadburn/vn5.8_accumulate_soil (last access: 16th July 2021, Chadburn 2021) which is a branch of JULESv5.8 with the new code described in this paper added. The runs were completed with the Rose suite https://code.metoffice.gov.uk/trac/roses-u/browser/c/g/3/6/5/, vn 200810 (last access: 16th July 2021, Burke and Chadburn 2021). Peat core data used for evaluation was derived during the "millipeat" project (UK Natural Research Council standard grant no. NE/1012915) and published in Gallego-Sala et al. (2018). The processed "millipeat" data that appears on the plots is available in the repository on Zenodo: https://doi.org/10.5281/zenodo.5549472, along with the full list of 696 DOI's that comprise the full dataset. All additional soil profile data used in this manuscript is also either provided or linked to from the Zenodo repository, https://doi.org/10.5281/zenodo.5549472. This repository further includes all of the JULES output data and all of the R code to recreate the plots in this manuscript using the JULES output data and observations (https://doi.org/10.5281/zenodo.5549472).

*Author contributions.*  SEC developed the model, performed simulations and analysis and wrote the first version of the manuscript. EJB set up the JULES suite and synthesised literature data. AVG provided observational data and expertise on peatland functioning. NDS contributed to the model development. MSB, DJC, JD, CWE, ESE, KF, YG, MN, WP, EAGS and SW provided model forcing and evaluation data. EJB, AVG, NDS, YG and SW contributed to the text of the manuscript.

*Competing interests.*  The authors declare no competing interests

*Acknowledgements.*  SEC was supported by a Natural Environment Research Council independent research fellowship (grant no. NE/R015791/1). AGS and DJC acknowledge funding from the UK Natural Research Council nos. NE/1012915 and NE/S001166/1. AGS receives funding from the European Research Council (ERC) under the European Union's Horizon 2020 research and innovation programme (grant agreement No 865403). This works reflects only the authors' view and the European Commission/Agency is not responsible for any use that may be made of the information it contains. EJB is supported by the Joint UK BEIS/Defra Met Office Hadley Centre Climate Programme (grant no. GA01101). The authors would like to thank Oliver Sonnentag, Dennis Baldocci, Julia Boike, Hanna Lee, David Walmsley, Han Dolman, Matthias Peichl, Mats Nilsson, Mika Aurela, Annalea Lohila, Christina Schaedel and Mathias Göckede for additional data and/or input.





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
