# Peer review of "A new approach to simulate peat accumulation, degradation and stability in a global land surface scheme (JULES vn5.8_accumulate_soil)"

_Geoscientific Model Development, 2021_

## Author Response (AR1)

**Reviewer 1 Comments**

The manuscript titled "A new approach to simulate peat accumulation, degredation and stability in global land surface scheme (JULES vn5.8_accumulate_soil)" introduces a new version of the JULES earth system model (ESM) which, as the title suggests, includes peat accumulation dynamics. The manuscript clearly demonstrates a substantial contribution to the science of modelling peat in ESMs. The authors describe a version of JULES that can track the vertical accumulation of peat coupled with thermal and hydraulic soil properties, which is novel for global-scale modelling. The authors do an excellent job of describing their methodology, including appropriate supplementary materials, as well as providing a link to the model code with what seems to be enough detailed information to reproduce the work. Their results were clearly and appropriately presented. The model is not performing well at the site-level for some site conditions, and I believe that some very minor adjustments to the discussion around this is required (see suggested revisions below). However, the model performs relatively well when spatially aggregated and is a major step towards improving the representation of peat in ESMs. I highly recommend that this manuscript is published after a few very minor revisions.

*We thank the reviewer for their positive comments!*

Suggested minor revisions:

L227. Authors refer to a 'dump' file here. I suggest the authors explain where the data came from in the dump file more clearly? In Table 2 there is one line in the caption, which states that the profiles came from Auchencorth but the rational for this was unclear and should be explained in the main text.

*We have rephrased the sentence that refers to a 'dump' file as follows:*

*"Each soil carbon pool in each soil layer is assigned an age, A, at the start of the simulation, which currently is either zero on initialising the spin-up or  can be initialised from a*n existing simulation *."*

*In order to explain why the Auchencorth profile was chosen to initialise simulations, we have added the following text:*

*"To initialise with peat we used the spun-up profile from Auchencorth JULES-Peat-W simulation, since this simulation had formed a thick, 1.7m layer of peat (see Figure S12). The only site that formed thicker peat in JULES-Peat was CA_WP1 which formed an extremely thick (5–6m) peat layer (Table S4, Figure S12), so we chose Auchencorth as a site with a thick but not extreme peat profile."*

L354. The authors conclude that "mineral soil carbon profiles can be adequately represented with all model versions", however in the previous paragraph and in Figure 2, it's clear that the model does not perform well for certain sites. The phrase is therefore misleading and should be re-phrased to reflect the appropriate scale at which the model can be applied (i.e. site-level accuracy still requires improvement, but all versions of the model perform well when aggregated/scaled-up).

*We have updated this following your suggestion:*

*"Overall we conclude that despite poor model performance at individual sites, the aggregated soil carbon profiles in both JULES and JULES-Peat adequately resemble observed mineral soil profiles (Figure 3h)."*

L385. The statement that "simulations without lateral water flow out of the soil compare best against observations is an indication that the model behavior is reasonable" did not make sense to me. I assume that the observations came from sites with lateral flow, so I'm not sure I understand

the point the authors are trying to make here. I understand that more peat should accumulate in areas with wetter conditions and less lateral flow, but I do not understand the link between the observational data made here.

*Apologies for the lack of clarity. This was indeed simply making the point that it's reasonable for more peat to develop in wetter areas, but just saying "observations" wasn't clear- I meant specifically against observations of peat. We have rephrased this sentence to:*

"*Since peat generally forms in wetter places, the fact that the simulations without lateral water flow out of the soil (-W and -W10) compare best against observations from peat soils is expected if the model realistically forms more peat in wetter soils.*"

**Reviewer 2 comments**

This manuscript is presenting a version of JULES that include a vertically discretize accumulation of organic and mineral soil. This version is also capable of simulating soil carbon aging. Capability of the soil organic accumulation scheme is demonstrated for various simulation set up that aim at showing sensitivity of the model to soil properties. Simulation results are compared with the standard JULES version and to peatland profiles at 216 sites. The paper also discusses the capability of the model to reproduce processes specific of peatlands formation, and stability. The novelty of this scheme is to be able to consider the resilience of peatland soil to increase atmospheric temperature and the instability of peatland soil to drainage. Thus, this manuscript represents a significant contribution to modelling science and is within the scope of Geoscientific Model Development. Model description (section 2) is well structured and the model is sorely described. Further than developing a new model for peatlands, this scheme has been designed for global scale simulations using Earth System Model (ESM). Therefore, I am convinced that the new features proposed in JULES-Peat will provide a complementary view on peatland functioning in the literature.

I have some comments that I think will help to ease the reading of the paper and I hope render the study even more appealing. Those modifications do not require to run some more simulations but aim at emphasis results that demonstrate the capability of the model to simulate resilience behaviors of peatland soil and the faster degradation of the peat soil due to a significant decrease in soil humidity.

I hope the authors find my suggestions useful and I am looking forward to read a revised version of the manuscript.

*We thank the reviewer for their positive comments and for the thorough review of the manuscript. We have addressed the comments as described below, and we do believe that this has improved the paper.*

General comments:

- The present land surface model JULES-peat in this manuscript is evaluated for tundra, boreal and temperate regions however there are no evaluation in tropical regions. Therefore, the title should be modified to mention that it is only for peat accumulation in the North hemisphere or also named northern peatlands. This should also be specified in the introduction and in the other parts of the manuscript.

*We have modified the title to specify that only high-latitude and temperate peatlands are included, so the title is now:*

"*A new approach to simulate peat accumulation, degradation and stability in a global land surface scheme (JULES vn5.8_accumulate_soil) for northern and temperate peatlands*"

*Similarly we added text in the abstract, introduction and discussion to specify that only northern and temperate sites are considered, and to discuss the need to address tropical peatlands in future:*

**Abstract:** *End of 2ⁿᵈ paragraph "The new scheme is tested and evaluated at northern and temperate sites."*

*"against typical peat profiles based on 216 northern and temperate sites from a global dataset of peat cores"*

**Introduction:** *"This scheme is implemented and demonstrated in the JULES land surface model, for northern and temperate sites."*

*Very end of section 4 (results and discussion), added: "Finally, the JULES-Peat model has not yet been tested in tropical peatlands, which differ from northern peatlands in terms of hydrology and vegetation, and have only recently gained attention in the modelling community (Kurnianto et al., 2015; Apers et al., 2021). There is a clear need for more focused study of tropical peatlands, given their large spatial extent and carbon stock, and the potential impacts of their ongoing drainage (Dargie et al., 2017; Mishra et al., 2021). Some of the key principles behind peat dynamics are universal (for example, suppression of decomposition in wet soils, dynamic growth of the soil surface), but model parameters such as those in the relationships used to determine soil characteristics may need to be updated for tropical peat (e.g. Equation 13–18)."*

*Conclusion: "As outlined above in Section 4.5, major challenges remain around appropriately modelling peatland vegetation and large-scale hydrology, as well as a need to test the model for tropical peatlands."*

- Section 3 present a series of simulations and the dataset that is employed for model evaluation. Almost all simulations defined in Table 2 are described in this section except for JULES-Peat-W-drain and JULES-Peat-W10-drain. Few sentences about these two simulations are needed to fully discuss Table 2 and the whole set of simulations.

*Thanks for pointing this out. We have added the following after discussing switching lateral flows off for JULES-PeatW(10): "In JULES-Peat-W-drain and JULES-Peat-W10-drain (Table 2), the lateral flow is set to zero during spin-up but switched back on during the main run to approximate drainage."*

- Section 4 describes and discusses simulation results. While results are very interesting the structure/organization of this section is a little confusing:

- I feel like those sections 4.1 and 4.2, are more like a results part and section 4.3 to 4.5 a discussion part. Therefore, I would recommend to split this section 4 results and discuss in two sections. The first one with the results described in sections 4.1 and 4.2 on the accumulation scheme and soil aging. The second section that will discuss results described in section 4.3, 4.4 and 4.5.

*We have split Section 4 into results and discussion as suggested.*

- At first, it has been surprising to me to start the result section of a model on peat (organic rich soil) by a paragraph on the representation of mineral soil. In addition, the following section is titles "JULES-Peat evaluation" which could mislead the reader in thinking that results in section 4.1 are not from JULES-Peat but only from JULES therefore current section 4.2 needs a different title.

*Since JULES is a global model and not just a model of peat, it is important that adding the functionality to represent peat soil does not degrade the performance at mineral soil sites. We added this at the start of the section on mineral soils to make this clear: "Since JULES is a global*

*model, it is important that adding the functionality to represent peat does not degrade model performance for mineral soils". We fully agree that "JULES-Peat evaluation" was a misleading title for following section, so we have renamed it to "Model evaluation at peatland sites" as well as dividing into two sections (see below).*

- Although, the authors should add a paragraph explaining how the different parts of this new scheme serves in representing vertical profiles of organic and mineral soils and their response to environment changes. For instance, how is the model performed a transition from mineral soil to organic soil or the other way around and which part(s) of the model is involved in this process. This paragraph can be placed either as introduction or as a sub-section of section 4: results and will aim at presenting the different components (refereeing to section 2 and equations) of the scheme. Emphasis needs to be made on the new functionality of the model. For example, line 407 to 412 can be moved to this new part.

*Thanks for the recommendation, we agree that these links were not made clear. As suggested we have added a new section to the results: 4.4 New processes in JULES-Peat. This contains the lines 407-412 as suggested and reads as follows:*

*"We initially introduced the additional processes and parameter changes that were incorporated in JULES-Peat into simulations one by one to test each process, before running the full model. These simulations are shown in the Supplementary material (Supplementary Section 2). The key development that allows the shape of the carbon profiles to be more realistic is accounting for the volume of organic material added and removed from the soil column (Equation 7) as described in Section 2.3. In particular, this process enables more carbon to reach the deeper soil, and makes the carbon density in the surface lower since it expands when plant litter (low density: $pdpmrpm = 35 kgm^{-3}$) is added. These differences are clear in all of the JULES-acc and JULES-Peat simulations in comparison to the original JULES (Figure S3 and Figure 5). The majority of the reduction in RMSE is already achieved by adding in this process alone (reduced from 23.0 $kgm^{-3}$ to 10.4 $kgm^{-3}$ at temperate sites, and 37.3 to 17.3 $kgm^{-3}$ for boreal sites; Table S2).*

*While it does not significantly reduce the RMSE by itself, modifying the moisture function to suppress decomposition when saturated (Section 2.2, Equation 3) allows more peat to form in wetter areas, which is a crucial factor in simulating realistic peatland distribution and future dynamics. Reducing drainage makes the simulation worse for mineral soil sites, which is exactly what would be expected (wetter soil → more peat forms), and it almost universally improves carbon profiles for organic sites (compare JULES-Peat-B and JULES-Peat-B10 against JULES-Peat-W and JULES-Peat-W10 in Table 3).*

*The other major new process introduced is that JULES-Peat simulates its own soil characteristics (Equations 13–18, Section 2.5). While we don't have measured profiles of soil characteristics to compare against, we can compare the soil thermal and hydraulic parameters simulated by JULES-Peat against those prescribed in JULES. Comparisons are shown in the supplementary material for key parameters: $K_{sat}$, $\Psi_{sat}$, $\theta_{sat}$ and b, for the simulations JULES-Peat-B, JULES-Peat-B10 and JULES-Peat-i10 (Supplementary Figures S7–S10). For some sites there is a good correspondence between the simulated and prescribed parameters, and others there are significant differences, but all simulated profiles behave sensibly.*

*Simulating these soil properties dynamically leads, in many cases, to a thicker organic layer (compare JULES-accC and JULES-accC10 with JULES-Peat-B and JULES-Peat-B10 in Table S4) and more soil organic carbon (Figures S5 and S6). This increase in carbon results in profiles that are significantly more realistic for organic sites (Figure S6; RMSE reduced from ~18–20 $kgm^{-3}$ to 13–14 $kgm^{-3}$) and marginally less realistic for mineral soil sites (Figure S5; RMSE increased from ~6–7 $kgm^{-3}$ to 6–8 $kgm^{-3}$). This suggests a self-reinforcing feedback of peat accumulation with*

*soil characteristics – e.g. if peat accumulation has started, it is more likely to continue – and can lead to various important dynamics which are discussed in the following sections (5.1 and 5.2)."*

- Results could be organized into more sub-sections such as "Evaluation of intrinsic conditions" that will include results on carbon profile, age and bulk density and "Evaluation of environmental conditions" that will include results on JULES-Peat-W and initial conditions.

*Thanks for the suggestion, we have divided the original section 4.2 (ambiguously named "JULES-Peat evaluation" in the original preprint) into the following two sections:*

*4.2 Model evaluation at peatland sites*

*4.3 Impact of environmental and initial conditions on peat profiles*

- In section 4.1, at the place of discussing the validation of each vertical profile, I would rather argue that in the top soil JULES-PEAT version provide more realistic carbon density. However, in soil layers deeper than 0.5m JULES-PEAT largely overestimate carbon content compare to JULES base case. Surprisingly while considering all the soil profile JULES_PEAT generate a carbon profile that fit better the observation. May be a RMSE or R2 will help quantifying the difference between observations and simulations?

*Thanks for the suggestion, we have now expanded the discussion and included RMSE values, as follows: "The lower carbon density at the surface matches better with observations than the original JULES simulation, but the carbon at depth tends to be overestimated. In terms of RMSE the aggregated profile is improved in JULES-Peat-B and JULES-Peat-B10 (RMSE 6.2 kgm−3 and 8.1 kgm−3) compared to JULES (RMSE 12.1 kgm−3); Table 3."*

- I like stand-alone figures it helps me capture the overall content and structure of a paper therefore additional information need to be added to all figure captions. Figure captions should define all the element of the figure in order for the reader to be able to understand the aim and content of the figure without relying on the main discussion in the manuscript (see details comments for each figure and table).

*Addressed in detailed comments, below.*

- It is sometimes difficult in the manuscript to distinguish between JULES the model and the simulation therefore the name of the simulation can me changed. It can be called after the reference number of the JULES version employed or JULES-BaseCase or JULES-ReferenceRun.

*Every simulation was performed with the same version of JULES (the branch vn5.8_accumulate_soil, which is referred to in the title and code availability section). It is standard that new functionality is added to JULES via switches which allow different functions in the model to be switched on or off when the model is run. Therefore, the "baseline" simulations are performed with the updated code but with the new functionality switched off, and every simulation shown in the paper is thus a different simulation with the same model. We have updated the text and the naming to make it clearer. In particular we explain early on that: "Note that all the simulations in this paper use the same branch of JULES. We generically refer to any configuration with the new peat functionality enabled as 'JULES-Peat', which is further sub-divided into different simulations." We have then re-named the specific sub-simulations that were confusingly labelled as "JULES-Peat" to "JULES-Peat-B" where B stands for baseline. We have been careful to make sure that we refer to configurations vs model versions in a correct and consistent way throughout the updated manuscript. For example on line 371 of the preprint we refer to "the original JULES" and have updated this to "the original JULES configuration".*

- In a few places, the authors are confusing ESM and land surface models. For instance, in line 7, 59 and 520 whereas in line 82 they explained that "JULES includes a vertically-resolved soil

carbon scheme (Burke et al., 2017a), although this hasn't yet been used in the Earth System Model configuration." Since this study was conducted on stand-alone version (offline conditions), ESMs line 7, 59 and 520 should be replaced by land surface model.

*Line 59: replaced.*

*Line 7: We argue that JULES is indeed an "ESM land surface scheme" since it is the land surface scheme of an earth system model (as well as its other uses), and therefore this is ok here since we wanted to highlight its role in an ESM.*

*Line 520: Similar to line 7.*

- Citations "in prep.":

I do think it is not appropriate to have among references papers in preparation e.i. Gao et al., in prep., Smith et al., in prep. In a manuscript in preparation many things can change from the reference authors, title, journal that later on, it gets difficult to figure which paper you were refereeing to. In addition, these references are not part of references listed at the end of the paper. Please remove these citations that appear in table 1, line 535 and line 290.

*Smith et al is no longer "in prep" and now has a DOI on GMDD, so I have added the citation for that (see below). The Gao et al paper is not yet submitted and available to cite, however I am optimistic that this will be the case by the time this paper would be fully published, so I will leave in the "in prep" citation in for now, and either replace it with a preprint citation or remove if necessary. We have added the "in prep" citation to the reference list for now.*

***Added references:** Smith, N. D., Chadburn, S. E., Burke, E. J., Schanke Aas, K., Althuizen, I. H. J., Boike, J., Christiansen, C. T., Etzelmüller, B., Friborg, T., Lee, H., Rumbold, H., Turton, R., and Westermann, S.: Explicitly modelling microtopography in permafrost landscapes in a land-surface model (JULES vn5.4_microtopography), Geoscientific Model Development Discussions, 2021, 1–43, https://doi.org/10.5194/gmd-2021-285, https://gmd.copernicus.org/preprints/gmd-2021-285/, 2021*

*Gao, Y. et al.: Multi-site evaluation of modelled methane emissions over northern wetlands by the JULES land surface model coupled with the HIMMELI peatland methane emission model, To be submitted to Sci. Total Environ., in prep.*

Detailed comments for figures and tables:

Figure 1: Panel F should be removed from this figure to be in a separate figure since you discuss figure 1F first in line 105 and 125 and later figure 1A-E line 183.

In panel 1F that should be the new figure 1, the legend is not very clear. It is a little confusing to have on one side line color and on the other line types. I recognize that it takes more room but it is also easier to understand having a legend like: green solid line JULES, wilt=0.1; green dashed line JULES, wilt=0.5; blue solid line JULES-peat, wilt=0.1; blue dashed line JULES-peat, wilt=0.1;

Then for the new figure 2 that will show panels A to E, only one legend for all panels can be displayed outside of panels or at the place of former panel F.

*Thanks for the suggestions. We agree that these figures should be separated. We have done so in the revised manuscript and moved the legends outside the figures as you suggest. We also updated the legend in the new Figure 1 as you suggested, it now looks like this:*

[Figure]

In the new figure captions, the vertical dotted lines or the region between dotted lines and the purple crosses 'compiled data' need to be defined. For example, line 265-267 can be placed in the figure caption to define the purple crosses.

*We have updated the caption to define both of these as requested: "Vertical dashed lines show the range of data that were used to fit the functions. These correspond to the minimum and maximum bulk densities for organic material that we derived for use in JULES ($\rho_{dpmrpm}$ and $\rho_{biohum}$; Section 2.3). The additional literature data for saturated hydraulic suction and Clapp-Hornberger exponent shown in purple were derived from the following papers: Londra (2010); Rezanezhad et al. (2012); Da Silva et al. (1993); Weiss et al. (1998); Päivänen et al. (1973); Boelter (1964); Rydén et al. (1980); Schwärzel et al. (2006)."*

Figure 2 and 3: There are some elements of the figure that have not be defined neither in the figure caption nor in the legend. For instance, crosses and diamond markers are not defined as well as in panel entitled "All", the black solid line and the grey area. Also, it will be easier for the reader whether in the figure caption there will be a brief description of the main difference between JULES-Peat and JULES-Peat10 such as at line 347-348 and of JULES-Peat-W and JULES-Peat-W10 such as in line362-363. I would prefer to have the legend outside of any panels so we can see the full vertical profile for Svalbard_Ny site even though it is easy to guess. It would be interesting to display in each panel the RMSD value of each site. These values will provide a quantitative evaluation of the fit for each simulation. It will be easier to refer to specific panels in each figure if each panel is labeled with a letter.

*We have now defined the markers, lines and shaded area in the captions as follows: "Observations are shown with black markers or lines. The black line on the final panel is the median of all sites, with the grey area indicating the interquartile range." We also added more details about the simulations, as requested. For example "In JULES-Peat-W and JULES-PeatW10 (dark blue and light blue lines) the lateral water flow is set to zero. The simulations with '10' on the end have $\tau_{lit}$ = 10". We also moved the legends to the outside of the figures as requested, and labelled each panel with a letter. We could not think of a good way to display RMSD values of each site, since there are up to 5 model runs per site, and we would have to specify which value is which, this would result in table with 5 numbers and simulation names associated with every single panel (as many as 9 panels per figure). However we have added quantitative values for the RMSE of the aggregated simulations in the text, both in reference to Figure 2 (now 3) – see earlier comment – and additionally for Figure 3 (now 4), where lines 360-362 of the preprint now read: "almost all sites are simulated significantly better in all of the JULES-Peat configurations than they are in JULES*

(RMSE of median profile *12.9–13.5 kgm⁻³ for JULES-Peat configurations shown, and 25.5 kgm⁻³ for JULES; * Table 3).”

Figure 4, 5 and 6: The grey area is not defined neither in the caption nor in the legend. Here the same, I would rather have a full view of the vertical profile for Tundra sites and have the legend outside of panels. In the panel title boreal and temperate are displayed in capital letter and no tundra, I guess all region name should be displayed in the same manner. If I understand correctly figure 4, for example each red line corresponds to the simulation using JULES at a different site within the considered region then site names should be given in the figure caption. Also, I do not understand why there are no simulation of JULES in the Tundra region?

*We have now defined the grey area in the figure captions (see response to the next comment). We also moved the legend outside of the plots, and made the capitalisation in the plot titles consistent. The lack of JULES data for the tundra region on these figures is because none of the tundra sites accumulated enough carbon in the JULES standard configuration to classify as peat. The plots only show sites with peat so that they are comparable against the peat core dataset. This is already explained in the text, but to make the plots more standalone we have added this in the Figure captions as well: "Only sites where the model simulates a sufficiently thick organic layer to classify as peat (>15% carbon by mass for ≥39cm, see Section 3) are shown". We decided it would be impractical to label every site for every simulation on these plots, since the list of sites that have peat for each model configuration is different, we would have to list ~5-8 sites six times over in the caption (i.e. around 40 site names). Instead we have added a new Supplementary Table S4. We state that "The sites that are shown for each simulation are highlighted in bold in Supplementary Table S4".*

 In the figure caption "against **typical** profile for peat cores" typical can be removed resulting in "against profile for peat cores".

*We have updated these captions, they now read: "against median profile from peat cores. The grey shaded area shows the interquartile range."*

Figure 9: In the figure caption, it should be explained why is there different line styles for the four panels on the left than for those on the right?

*This was a mistake, we have made all lines solid in the updated manuscript.*

Figure 10: May be "Demonstration of peat accumulating / not accumulating" could be replace by "Demonstration of peat accumulation efficiency". The horizontal solid line in all four bottom panel needs to be defined somehow.

*Thanks for the suggestion. We have rephrased this to "Demonstration of how peat accumulation rate can depend on initial conditions" and removed the horizontal lines since they were not supposed to be on the plot.*

Table 4: There is no need of the second column "Colour in Fig.7" since there is a legend in Figure 7 that already define that. - *Removed.*

Detailed comments for the manuscript:

Line17-21: "peatlands. In particular the best performing configurations had root mean squared error (RMSE) in carbon density for peat sites of 7.7–16.7 kgC m−3 depending on climate zone, when compared against typical peat profiles based on 216 sites from a global dataset of peat cores. This error is considerably smaller than the soil carbon itself (around 30–60 kgC m−3) and reduced by 35–80% compared with standard JULES. The RMSE at mineral soil sites is also smaller in JULES-Peat than JULES itself (reduced by 30–50%)." This part is not clear when it is read for the first time

*We have re-written this for clarity, it now reads: "We evaluate the model against typical peat profiles based on 216 northern and temperate sites from a global dataset of peat cores. The root mean squared error in soil carbon profile is reduced by 35–80% in the best-performing JULES-Peat simulations compared with the standard JULES configuration. The RMSE in these JULES-Peat simulations is 7.7–16.7 kgC m$^{-3}$ depending on climate zone, which is considerably smaller than the soil carbon itself (around 30–60 kgC m$^{-3}$). The RMSE at mineral soil sites is also reduced in JULES-Peat compared with the original JULES configuration (reduced by ~30–50%)."*

Line 23-24: "This provides a new approach for improving the simulation of organic and peatland soils, and associated carbon-cycle feedbacks in ESMs, which other land surface models could follow." I believe that it is wiser to let the reader decide whether or not he want to use the same or a different implementation. I would remove the last part of the sentence "which other land surface models could follow"

*We have deleted this as suggested.*

Line 90-94: "These changes were made based on well-known principles: Firstly, that microbial activity drops to zero in completely dry conditions (Yan et al., 2018); secondly that respiration in anaerobic conditions is known to be no higher than 20% of the maximum rate in aerobic conditions (Schuur et al., 2015); and finally that when microbes lack nitrogen, they tend to decompose plant litter faster in order to 'mine' for nitrogen (Craine et al., 2007) in contrast to the original scheme introduced by Wiltshire et al. (2021) in which the decomposition of litter is inhibited when nitrogen is in short supply. " This sentence is very long and can be split in multiple sentences by removing the semi-column and word "that" after firstly, secondly and finally.

*Thanks for the suggestion, we have made these changes.*

Line 95: "as follow:" can be removed. *We believe the sentence is clearer with this in so we have left it as it is for now.*

Line 165:" See original JULES version on Figure 4, red lines" please add a reference or "Figure 4 of the present study" if it is what you mean. *We have updated this to "Figure 4 of this study"*

Line 202: "if there is a '**corner'** in the function" I believe that it is name a 'discontinuity' in a function rather than a 'corner'.

*We agree that 'corner' is an ambiguous word. However 'discontinuity' could imply that there is a 'gap' in the function. We have instead rephrased to "if the gradient of the function changes sharply", which we believe is sufficiently accurate and clear.*

Line 260: "firstly" can be removed. *Done*

Line290-292: "along with Scotty Creek (Helbig et al., 2016, 2017a, b), Pleistocene Park (Euskirchen et al., 2017b), Imnavait (Euskirchen et al., 2017a), and Eight Mile Lake (Celis et al., 2019)." Why did you add these sites for the present study? Are they more recent measurements? Are they complementary of your initial sites? If yes in which way? Do they add some representation of boreal or tundra sites?

*We simply used as many sites as we had available and processed data. The new sites were included as the site PI's had recently shared their data with us and we had processed this into the format we need for JULES. We have added the following to clarify this:*

*"We were able to include the four new sites that have not been used in previous studies with JULES due to additional data becoming available."*

Line 293: "Some of the sites, namely Abisko, Seida and Imnavait, **are split into** different landscape types," ; "are split into" can be replaced by "have".

*To make this clearer, we replaced "are split into" with "provided data from", since other sites may have different landscape types but we are only able to do multiple simulations for these specific sites that provided information on the difference between land cover types at the site. The sentence now reads: "Some of the sites, namely Abisko, Seida and Imnavait, provided data from different landscape types, resulting in 29 simulations in total."*

Line 303: "layered soil carbon and nitrogen are **switched on**; a bedrock column is included below the soil to simulate heat conduction. Starting from this baseline simulation, we then **switched on** the new processes in JULES-Peat"; the first "switched on" can be replaced by "considered" and the second one by "activated".

*Thanks for this. We made the second replacement as recommended, and changed the first "switched on" to "simulated" for extra clarity, as it is not clear that a model considers something.*

Line 321: ", or in other words more of the plant litter added to the surface layers." can be remove, it is already explained right above line 319.

*Deleted*

Line 327: "the fraction of organic matter that is carbon" can be replace by "the fraction carbon in soil organic matter" And "For this calculation we **required**" can be replaced by "selected". *See next comment*

Line 328-330: "the datapoints that were **essentially organic material** with minimal mineral content, **so we removed any datapoints** for which the percent carbon (by mass) was less than 30%, leaving only data where the vast majority of the soil by volume is organic material. This left over 24,500 datapoints. "can be replace by "the datapoints that were **organic rich** with minimal mineral content, for which the percent carbon (by mass) was **higher** than 30%, leaving only data**, over 24,500 datapoints,** where the vast majority of the soil by volume is organic material."

*Thanks, we have incorporated these suggestions, and the ones above, except we split the last sentence into two. The text now reads:*

*"We also used this dataset to derive the values for maximum and minimum bulk density (ρdpmrpm and ρbiohum, Section 2.3) and the fraction of carbon in organic matter (fc). For this calculation we selected only the datapoints that were organic rich with minimal mineral content, for which the percent carbon (by mass) was higher than 30%. This left over 24,500 datapoints where the vast majority of the soil by volume is organic material."*

Line 378: "32% and 56% for the configurations that perform best in terms of carbon profile (Table 3)." Could you name the configurations that performed best?

*We have added this. The text now reads:*

*"RMSE in age is reduced by 32% (temperate sites) and 56% (boreal sites) for the configurations that perform best in terms of carbon profile (JULES-Peat-W and JULES-Peat10 respectively, Table 3)."*

Line 385: "Since peat generally forms in wetter places, the fact that the simulations without lateral water flow out of the soil (-W and -W10) compare best against observations is an indication that the model behaviour is reasonable." Could it be an indication that the hydrology module is not well calibrated or that some processes are missing for peatland soil?

*Apologies this was not clear. We were just trying to make the point that more peat forms when the model is wetter, as expected. We addressed this in the reply to reviewer 1, above. Indeed the hydrology still needs more development as discussed in Section 4.5 (now 5.3 in revised manuscript). The sentence that you mentioned now reads: "Since peat generally forms in wetter*

*places, the fact that the simulations without lateral water flow out of the soil (-W and -W10) compare best against observations from peat soils is expected if the model realistically forms more peat in wetter soils."*

Line 392: "JULES-Peat was **only** able to accumulate peat **from scratch** at one of the tundra sites" can be modified to "JULES-Peat was **sometimes** able to accumulate peat **starting with no carbon in the soil at t=0** at one of the tundra sites"

*Thanks for the suggestion – done.*

Line 422: "Liu et al. (2020) tracked the surface subsidence rate over time following drainage in two different ecosystem types – forest and agriculture." Please explain a little more this study my understanding of this sentence was that Liu et al studied the subsidence of forest and agricultural lands and not of peatlands.

*Apologies that this was not clear. These were in fact peatlands, which were drained for forestry and agriculture. We have clarified this as follows:*

*"Liu et al. (2020) tracked the surface subsidence rate over time following drainage of peatlands for two different land use practices in two different ecosystem types - forestry and agriculture."*

Line 427: "the mechanical raising and lowering of the peat surface" Could you provide the order of magnitude of heigh change during bog breathing process?

*We have added this: "the mechanical raising and lowering of the peat surface by as much as tens of cm as the water table fluctuates"*

Line 447: "It is significant that the UK **are** Ireland sites" I think "are" need to be replace by "and".

*Thanks – done.*

Supplementary information:

In equation 1 and 4: does dz has the same definition than €€dz = z-zeff or is it dzeff?

*dz is a derivative and 'z' is the variable. We apologise for also naming a specific value of 'z' as 'z', that was indeed very confusing! Thank you for picking this up. We have renamed the point 'z' as 'z$_d$' which is what it's already called on the diagram, and defined z explicitly. ("z is vertical distance (m)")*

In section 1: "The€€first order derivative (second term on **RHS** of Equation 1)" please defined RHS.

*Done*

For all figures, I would recommend to place the legend outside of any panels for example above or below the figure. For Figure S7 to S10, please add the legend, it is quite annoying when you are looking at figure S10 and that you have to scroll back 3 or 4 pages to check the legend in Figure S5 and S6.
*We have now placed the legends outside of the figures for all supplementary figures (except for S11 where it is placed in the bottom right, far from any data points), and added colour legends on Figures S7-S10 as requested.*